# Leveraging Generative Trajectory Mismatch for Cross-Domain Policy Adaptation

## Abstract

Transferring policies across domains poses a vital challenge in reinforcement learning, due to the dynamics mismatch between the source and target domains. In this paper, we consider the setting of online dynamics adaptation, where policies are trained in the source domain with sufficient data, while only limited interactions with the target domain are allowed. There are a few existing works that address the dynamics mismatch by employing domain classifiers, value-guided data filtering, or representation learning. Instead, we study the domain adaptation problem from a generative modeling perspective. Specifically, we introduce DADiff, a diffusion-based framework that leverages the discrepancy between source and target domain generative trajectories in the generation process of the next state to estimate the dynamics mismatch. Both reward modification and data selection variants are developed to adapt the policy to the target domain. We also provide a theoretical analysis to show that the performance difference of a given policy between the two domains is bounded by the generative trajectory deviation. More discussions on the applicability of the variants and the connection between our theoretical analysis and the prior work are further provided. We conduct extensive experiments in environments with kinematic and morphology shifts to validate the effectiveness of our method. The results demonstrate that our method provides superior performance compared to existing approaches, effectively addressing the dynamics mismatch. We provide the code of our method at `https://anonymous.4open.science/r/DADiff-release-83D5`.

## 1 Introduction

Reinforcement learning (RL) has shown strong potential in complex decision-making tasks, but training directly in the real-world environment (*target domain*) is often restricted by safety, cost, and limited interaction budgets. An alternative strategy is to train policies in a surrogate environment (*source domain*), such as a simulator, and then transfer them to the target domain. But due to the dynamics mismatch between the source and target domains, directly transferring the policy often leads to performance degradation, which is a critical challenge in the sim-to-real problem (Zhao et al., 2020; Da et al., 2025). One solution to this transfer problem is known as *online dynamics adaptation* (Xu et al., 2023; Lyu et al., 2024b), where policies are trained with abundant source-domain data and only limited interactions in the target domain. In this setting, the state space, action space, and reward function remain consistent across domains, while the transition dynamics differ. Compared with solutions such as domain randomization (Peng et al., 2018; Mehta et al., 2020; Curtis et al., 2025) or simulator calibration (Chebotar et al., 2019), online dynamics adaptation does not require access to high-fidelity simulators or prior knowledge of target dynamics, and can therefore be applied in situations where such information is unavailable.

Existing online dynamics adaptation methods, including classifier-based approaches (Eysenbach et al., 2021), value-guided filtering (Xu et al., 2023), and representation learning (Lyu et al., 2024a), capture dynamics discrepancy from different perspectives: classifiers provide coarse distinctions between domains, value-guided methods depend on the modeling of forward predictions, and representation learning relies on assumptions of invariant latent structures across domains. When the domains are complex or stochastic, a key challenge that remains is to develop an approach capable of capturing dynamics discrepancy in a more fine-grained and distributional manner.

The generative modeling perspective provides a potential direction. Generative models, such as diffusion models (Sohl-Dickstein et al., 2015; Ho et al., 2020; Song et al., 2021) and flow matching methods (Lipman et al., 2022; Liu et al., 2023), have demonstrated strong capability in representing complex distributions. When state transitions are viewed as a conditional generative process, the mismatch between source and target domains can be interpreted as a discrepancy between their respective generative processes. Specifically, the multi-step sampling procedure in diffusion models and flow matching methods produces several latent states, which construct a generative trajectory, serving as structured signals of source–target dynamics deviation. These latent states allow the discrepancy to be captured not only at the next-state level but also along the entire trajectory. Intuitively, if the source and target domains follow different dynamics, their trajectories will diverge at multiple steps, a phenomenon we term *generative trajectory deviation*. This notion provides a fine-grained view of dynamics discrepancy by revealing how divergence accumulates along the trajectory, rather than relying solely on local or aggregated comparisons. Our theoretical analysis further connects trajectory deviation to performance guarantees, providing motivation for algorithmic design.

Building on this perspective, we introduce **DADiff**, a diffusion-based framework for online dynamics adaptation. DADiff leverages latent states in diffusion models to measure generative trajectory deviation between source and target domains, and exploits this deviation in two complementary ways: (i) **DADiff-modify**, which adjusts source-domain rewards with deviation-based penalties, and (ii) **DADiff-select**, which filters source-domain data based on deviation before value function updates. We further discuss the applicability of these variants to different tasks, highlight the advantages of our method compared to prior work, and establish a connection between our analysis and the theoretical guarantee of prior work. Empirical results in environments with kinematic and morphology shifts show the superior performance of our method compared to existing algorithms.

## 2 RELATED WORKS

**Domain Adaptation in RL**  Generalizing RL policies to diverse environments is critical for real-world deployment, where transition dynamics (Eysenbach et al., 2021; Viano et al., 2021; Xue et al., 2023; Da et al., 2024), state or action spaces (Gamrian & Goldberg, 2019; Ge et al., 2023; Heng et al., 2022; Pan et al., 2025) may be different. To address domain adaptation, prior work falls under three categories: (i) domain randomization that randomizes transition dynamics to expose agents to many environment configurations (Slaoui et al., 2019; Mehta et al., 2020; Vuong et al., 2019; Jiang et al., 2024), (ii) meta-learning to few-shot adapt to many environments (Nagabandi et al., 2018; Arndt et al., 2020; Wu et al., 2022), and (iii) expert demonstrations of target environments through imitation learning (Raychaudhuri et al., 2021; Fickinger et al., 2022). However, these approaches are either computationally expensive (meta-learning) or require hard-to-obtain demonstrations (imitation learning). With only limited target-domain data, some works perform reward modifications to transition to the target domain by using transition classifiers (Eysenbach et al., 2021; Guo et al., 2024) or reward augmentations (Van et al., 2024; Lyu et al., 2024b). Data selection methods (Xu et al., 2023; Wen et al., 2024) have also been used to filter out part of the source-domain transitions and train policies on both source and target domain data. When the domains are complex or stochastic, a key challenge that remains is to develop an approach capable of capturing the dynamics discrepancy. Our method explores this challenge from a generative modeling perspective by measuring the generative trajectory deviation between the source and target domains.

**Diffusion Models in RL**  Diffusion models (Sohl-Dickstein et al., 2015; Ho et al., 2020; Song et al., 2021) have been extensively used for generating effective decision-making policies in several domains, such as RL (Kang et al., 2023), robotics (Chi et al., 2023), and planning (Janner et al., 2022). Specifically, they are widely leveraged to synthesize data for offline RL (Lu et al., 2023), facilitate planning and action generation in multi-task scenarios (He et al., 2023), and enhance the representational capacity of learned RL policies (Wang et al., 2024). In addition, diffusion models have also been extended to the multi-agent settings (Zhu et al., 2024) and for hierarchical RL (Li et al., 2023). In the field of domain adaptation, they are utilized to augment the target-domain data in order to boost the performance of offline RL policies (Van et al., 2025). However, the introduction of synthesizers may lead to extra computational costs, and the quality of synthesized data is hard to guarantee. In contrast, we choose to directly estimate the dynamics discrepancy by multiple latent states from diffusion models instead of generating more synthetic data.

## 3 PRELIMINARIES

**Online Dynamics Adaptation**  We consider two Markov Decision Processes (MDPs), denoted as $\mathcal{M}_{\text{src}} = (\mathcal{S}, \mathcal{A}, P_{\text{src}}, r, \gamma)$ and $\mathcal{M}_{\text{tar}} = (\mathcal{S}, \mathcal{A}, P_{\text{tar}}, r, \gamma)$ for the source domain and target domain, respectively. The state space $\mathcal{S}$, action space $\mathcal{A}$, reward function $r : \mathcal{S} \times \mathcal{A} \to \mathbb{R}$ and discount factor $\gamma \in [0, 1]$ are consistent across both domains, while the transition dynamics $P_{\text{src}}$ and $P_{\text{tar}}$ differ. The goal of online dynamics adaptation is to learn a policy $\pi$ that achieves high performance in the target domain $\mathcal{M}_{\text{tar}}$, utilizing sufficient data from the source domain and only limited interactions from the target domain. In addition, we specify a domain $\mathcal{M}$ and define the probability that a policy $\pi$ encounters a state $s$ at time step $t$ as $P_{\mathcal{M},t}^\pi(s)$. Therefore, the normalized probability that a policy $\pi$ visits a state-action pair $(s, a)$ in the domain $\mathcal{M}$ can be represented as $\rho_{\mathcal{M}}^\pi(s, a) :=$ $(1 - \gamma) \sum_{t=0}^{\infty} \gamma^t P_{\mathcal{M},t}^\pi(s)\pi(a|s)$. The expected return of a policy $\pi$ in $\mathcal{M}$ is defined as $\eta_{\mathcal{M}}(\pi) =$ $\mathbb{E}_{(s,a) \sim \rho_{\mathcal{M}}^\pi}[r(s, a)]$. We assume the reward are bounded by $|r(s, a)| \leq r_{\max}, \forall s \in \mathcal{S}, a \in \mathcal{A}$.

**Diffusion Models**  Diffusion models (Sohl-Dickstein et al., 2015; Ho et al., 2020; Song et al., 2021) are a family of generative models that learn to generate samples from a target distribution. We mainly focus on the denoising diffusion probabilistic model (DDPM) (Ho et al., 2020) in this paper. DDPM consists of a forward process and a reverse process. The forward process is regarded as a Markov chain that gradually adds noise to data, transforming a clean data point $x_0$ into Gaussian noise, which is formulated as follows,

$$x_k = \sqrt{1 - \beta_k}x_{k-1} + \sqrt{\beta_k}\epsilon, \quad \epsilon \sim \mathcal{N}(0, I), \tag{1}$$

where $x_k$ is the noisy data at diffusion timestep $k$, $\beta_k$ is the noise schedule, and $\epsilon$ is Gaussian noise. To simplify the forward process, we can directly sample the noisy data at diffusion timestep $k$ as follows,

$$x_k = \sqrt{\bar{\alpha}_k}x_0 + \sqrt{1 - \bar{\alpha}_k}\epsilon, \quad \epsilon \sim \mathcal{N}(0, I), \tag{2}$$

where $\alpha_k = 1 - \beta_k$ and $\bar{\alpha}_k = \prod_{i=1}^{k} \alpha_i$. The reverse process learns to denoise the noisy data step by step, which is formulated as follows,

$$x_{k-1} = \frac{1}{\sqrt{\alpha_k}}(x_k - \frac{\beta_k}{\sqrt{1 - \bar{\alpha}_k}}\epsilon_\theta(x_k, k)) + \sqrt{\frac{1 - \bar{\alpha}_{k-1}}{1 - \bar{\alpha}_k}\beta_k}\epsilon, \quad \epsilon \sim \mathcal{N}(0, I), \tag{3}$$

where $\epsilon_\theta(x_k, k)$ is a noise model that estimates the noise from the noisy data point $x_k$. The noisy data points $\{x_k\}_{k=0}^{K}$ form a generative trajectory from the initial noisy data $x_K$ to the clean data $x_0$. The training objective of the noise model is formulated as follows,

$$\mathcal{L}_{\text{diff}} = \mathbb{E}_{x_0,\epsilon,k} \left[ ||\epsilon - \epsilon_\theta(\sqrt{\bar{\alpha}_k}x_0 + \sqrt{1 - \bar{\alpha}_k}\epsilon, k)||^2 \right]. \tag{4}$$

## 4 METHODOLOGY

In this section, we first introduce a theoretical analysis to demonstrate the connection between the dynamics mismatch and the generative trajectory mismatch. Then, we present our diffusion-based method, DADiff, which measures the generative trajectory deviation from the perspective of diffusion models and adapts the learned policy to the target domain. The overview of our method is shown in Figure 1.

### 4.1 THEORETICAL ANALYSIS

Before introducing the theoretical analysis, we first provide the definition of a generative trajectory, which is crucial for the analysis. For clarity, we denote the next state $s'$ as $s_0'$.

**Definition 4.1 (Generative trajectory.)** Specify a domain $\mathcal{M}$ with transition dynamics $P_{\mathcal{M}}(s_0'|s, a)$. There is a generative trajectory for the next state $s_0'$ consisting of $K$ auxiliary variables $\{s_k'\}_{k=1}^{K}$, referred to as latent states. These latent states form a Markov chain from the initial latent state $s_K'$ to the next state $s_0'$ conditioned on the state–action pair $(s, a)$.

*Remark.* The Markov-chain definition enables the transition dynamics to be decomposed into multiple conditional probabilities, *i.e.*, $P_{\mathcal{M}}(s_0'|s, a) = \int P_{\mathcal{M}}(s_K'|s, a) \prod_{k=1}^{K} P_{\mathcal{M}}(s_{k-1}'|s_k', s, a)ds_{1:K}'$.

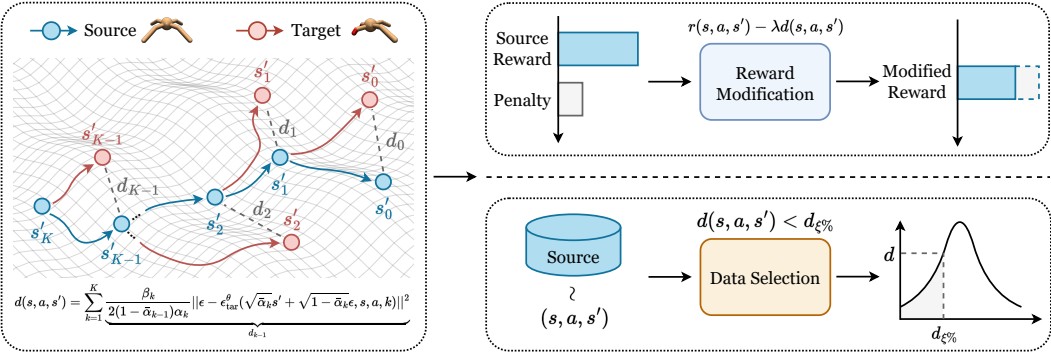

Figure 1: Illustration of DADiff. The left part visualizes the generative trajectories in the source and target domains. The deviation $d(s, a, s')$ is measured by the discrepancy $d_k$ of each latent state $s'_k$ in the source and target domain generative trajectories. The right part shows two ways to utilize the deviation $d(s, a, s')$ to adapt the policy to the target domain, *i.e.*, penalizing the source domain rewards (top right) or filtering source domain transitions (bottom right). The downstream SAC algorithm is then updated with both source and target domain data.

In this way, the next state $s'_0$ can be viewed as being generated step by step with latent states, forming a generative trajectory. The discrepancy of such generative trajectories across domains provides a natural estimation of the dynamics discrepancy.

We construct generative trajectories in both source and target domains, starting from the same initial latent state $s'_K$, and derive Theorem 4.2 to establish the connection between the dynamics mismatch and the generative trajectory mismatch. The detailed proof is provided in Appendix B.2.

**Theorem 4.2 (Performance bound controlled by generative trajectory discrepancy.)** *Denote* $\mathcal{M}_{\mathrm{src}}$ *and* $\mathcal{M}_{\mathrm{tar}}$ *as the source and target domains with different dynamics, respectively. The performance difference of any policy* $\pi$ *evaluated in* $\mathcal{M}_{\mathrm{src}}$ *and* $\mathcal{M}_{\mathrm{tar}}$ *can be bounded as below,*

$$\eta_{\mathcal{M}_{\mathrm{src}}}(\pi) - \eta_{\mathcal{M}_{\mathrm{tar}}}(\pi) \leq \frac{\sqrt{2}\gamma r_{\max}}{(1-\gamma)^2} \underbrace{\mathbb{E}_{\rho_{\mathrm{src}}^{\pi}}\left[\sqrt{\mathbb{E}_{P_{\mathrm{src}}}\left[D_{\mathrm{KL}}(P_{\mathrm{src}}(s'_K|s,a)||P_{\mathrm{tar}}(s'_K|s,a))\right]}\right]}_{(a): \text{ initial latent state deviation}}$$

$$+ \frac{\sqrt{2}\gamma r_{\max}}{(1-\gamma)^2}\mathbb{E}_{\rho_{\mathrm{src}}^{\pi}}\left[\sqrt{\mathbb{E}_{P_{\mathrm{src}}}\left[\sum_{k=1}^{K} D_{\mathrm{KL}}(P_{\mathrm{src}}(s'_{k-1}|s'_k,s,a)||P_{\mathrm{tar}}(s'_{k-1}|s'_k,s,a))\right]}\right] \quad (5)$$

$$\underbrace{\phantom{+ \frac{\sqrt{2}\gamma r_{\max}}{(1-\gamma)^2}\mathbb{E}_{\rho_{\mathrm{src}}^{\pi}}\left[\sqrt{\mathbb{E}_{P_{\mathrm{src}}}\left[\sum_{k=1}^{K} D_{\mathrm{KL}}\right]}\right]}}_{(b): \text{ latent state transition mismatch}}$$

***Remark.*** This bound indicates that the performance difference of a policy $\pi$ between the source and target domains is controlled by the initial latent state deviation term (a) and the latent state transition mismatch term (b). Since the generative trajectories in both the source and target domains share the same initial latent state $s'_K$, term (a) vanishes, leaving term (b) as the sole determinant of the performance difference. In other words, as long as the generative trajectories are similar in the source and target domains, the performance difference is small, and vice versa. We note that PAR (Lyu et al., 2024a) can be considered as a special case of Theorem 4.2 when $K = 1$. A discussion on the connection between our analysis and the theoretical guarantee of PAR is provided in Section 6.

### 4.2 DOMAIN ADAPTATION WITH DIFFUSION

Theorem 4.2 provides a theoretical guarantee linking the performance difference of a policy $\pi$ to the generative trajectory, thereby motivating a careful design of latent states in the trajectory. Since latent states are auxiliary constructs for capturing dynamics mismatch, the distribution of latent state transitions is not fixed and can be defined in different ways. In this section, we adopt the formulation of DDPM as an example to better characterize the dynamics discrepancy. In addition,

another implementation based on flow matching (Lipman et al., 2022; Liu et al., 2023) is provided in Appendix C.

We first redeclare the reverse process of DDPM in a reparameterized form to describe the latent state transition in domain $\mathcal{M}$ as follows,

$$s'_{k-1} = \frac{1}{\sqrt{\alpha_k}}(s'_k - \frac{\beta_k}{\sqrt{1 - \bar{\alpha}_k}}\epsilon_{\mathcal{M}}(s'_k, s, a, k)) + \sqrt{\frac{1 - \bar{\alpha}_{k-1}}{1 - \bar{\alpha}_k}\beta_k}\epsilon, \quad \epsilon \sim \mathcal{N}(0, I), \quad (6)$$

where $\epsilon_{\mathcal{M}}(s'_k, s, a, k)$ is the noise from the latent state $s'_k$ in domain $\mathcal{M}$. It indicates that the latent state transition follows a Gaussian distribution, *i.e.*,

$$P_{\mathcal{M}}(s'_{k-1}|s'_k, s, a) \sim \mathcal{N}(\frac{1}{\sqrt{\alpha_k}}(s'_k - \frac{\beta_k}{\sqrt{1 - \bar{\alpha}_k}}\epsilon_{\mathcal{M}}(s'_k, s, a, k)), \frac{1 - \bar{\alpha}_{k-1}}{1 - \bar{\alpha}_k}\beta_k I). \quad (7)$$

According to Theorem 4.2, the performance difference of a policy $\pi$ across domains is determined by the latent state transition mismatch term (b). Therefore, we can estimate the generative trajectory deviation $d(s, a, s')$ with the defined distribution of latent state transition in Equation 7 as follows,

$$d(s, a, s') = \sum_{k=1}^{K} D_{\text{KL}}(P_{\text{src}}(s'_{k-1}|s'_k, s, a)||P_{\text{tar}}(s'_{k-1}|s'_k, s, a))$$

$$= \sum_{k=1}^{K} \frac{\beta_k}{2(1 - \bar{\alpha}_{k-1})\alpha_k} \left\| \epsilon_{\text{src}}(s'_k, s, a, k) - \epsilon_{\text{tar}}(s'_k, s, a, k) \right\|^2. \quad (8)$$

We derive this equation by applying Lemma B.2 to compute the KL divergence between two Gaussian distributions. Notably, as the state transition tuple $(s, a, s')$ comes from the source domain, the noise $\epsilon_{\text{src}}(s'_k, s, a, k)$ estimated in the reverse process must be consistent with the noise used in the forward process to generate the latent state $s'_k$, which indicates $\epsilon_{\text{src}}(s'_k, s, a, k) = \epsilon$ with $\epsilon \sim \mathcal{N}(0, I)$. Besides, we introduce a noise model $\epsilon^\theta_{\text{tar}}(s'_k, s, a, k)$, trained with target-domain data, to estimate the noise in the target domain. The training objective is formulated as follows,

$$\mathcal{L}_{\text{noise}} = \mathbb{E}_{(s,a,s') \sim \mathcal{D}_{\text{tar}}, \epsilon, k} \left[ \left\| \epsilon - \epsilon^\theta_{\text{tar}}(\sqrt{\bar{\alpha}_k}s'_0 + \sqrt{1 - \bar{\alpha}_k}\epsilon, s, a, k) \right\|^2 \right]. \quad (9)$$

This objective mirrors the standard DDPM training loss, but conditions on $(s, a)$ to capture dynamics in the target domain. For the latent state $s'_k$ in Equation 8, there are two ways to obtain it: (i) by iteratively applying the reverse process in Equation 6, and (ii) by sampling directly from the forward process of DDPM, *i.e.*, $s'_k = \sqrt{\bar{\alpha}_k}s'_0 + \sqrt{1 - \bar{\alpha}_k}\epsilon$ with $\epsilon \sim \mathcal{N}(0, I)$. Specifically, the first way requires sequential sampling across all steps to generate the entire generative trajectory, which is computationally expensive. In contrast, the second way can produce all latent states in parallel, yielding a much more efficient implementation. Therefore, we choose to obtain the latent state $s'_k$ via the forward process in our method. We provide a visualization to compare these two ways for better understanding in Figure 7, Appendix E.2. Finally, the deviation $d(s, a, s')$ can be practically estimated as follows,

$$d(s, a, s') = \sum_{k=1}^{K} \frac{\beta_k}{2(1 - \bar{\alpha}_{k-1})\alpha_k} \left\| \epsilon - \epsilon^\theta_{\text{tar}}(\sqrt{\bar{\alpha}_k}s'_0 + \sqrt{1 - \bar{\alpha}_k}\epsilon, s, a, k) \right\|^2, \quad \epsilon \sim \mathcal{N}(0, I). \quad (10)$$

We further introduce two variants based on SAC (Haarnoja et al., 2018) to utilize the deviation $d(s, a, s')$, including reward modification and data selection, since we find that baselines adopting these two techniques exhibit complementary advantages in different tasks, which is shown in Section 5.2. We analyze the possible reason for this phenomenon from the reward distribution aspect in Section 6. The details of DADiff variants are provided as follows.

**Reward modification.** We refer to this variant as DADiff-modify. It adopts the deviation $d(s, a, s')$ as a reward penalty to modify the reward function in the source domain, *i.e.*,

$$r_{\text{mod}}(s, a, s') = r(s, a, s') - \lambda d(s, a, s'), \quad (11)$$

where $\lambda$ is a penalty coefficient to balance the original reward and the penalty. The objective function for training the value function gives,

$$\mathcal{L}_{\text{critic}} = \mathbb{E}_{(s,a,r_{\text{mod}},s') \sim \mathcal{D}_{\text{src}} \cup \mathcal{D}_{\text{tar}}} \left[ (Q_\phi - \mathcal{T}Q_\phi)^2 \right], \quad (12)$$

where $\mathcal{D}_{\text{tar}}$ and $\mathcal{D}_{\text{src}}$ are the datasets from the target and source domains, respectively, $Q_\phi$ is the value function, and $\mathcal{T}$ is the Bellman operator.

**Data selection.** We refer to this variant as DADiff-select. We select fixed percentage data with the lowest deviation $d(s, a, s')$ from a batch of source domain data. The selected data is then used to update the value function. We formulate the objective function of the value function as follows,

$$\mathcal{L}_{\text{critic}} = \mathbb{E}_{(s,a,r,s')\sim\mathcal{D}_{\text{tar}}} \left[(Q_\phi - \mathcal{T}Q_\phi)^2\right] + \mathbb{E}_{(s,a,r,s')\sim\mathcal{D}_{\text{src}}} \left[\omega(s, a, s')(Q_\phi - \mathcal{T}Q_\phi)^2\right], \quad (13)$$

where $\omega(s, a, s') = \mathbb{1}(d(s, a, s') < d_{\xi\%})$, $\mathbb{1}$ is the indicator function, and $d_{\xi\%}$ denotes the lowest $\xi$-quantile deviation in the batch.

For both variants, the objective function of the policy $\pi$ is formulated as:

$$\mathcal{L}_{\text{actor}} = \mathbb{E}_{(s,a,r,s')\sim\mathcal{D}_{\text{src}}\cup\mathcal{D}_{\text{tar}}} \left[-\min_{i=1,2} Q_{\phi_i}(s, a) + \tau \log \pi(a|s)\right], \quad (14)$$

where $\tau$ is the entropy temperature coefficient, and $i$ denotes the value function index. We provide the pseudocode of DADiff in Algorithm 1, Appendix D.

## 5 EXPERIMENTS

In this section, we conduct experiments to evaluate the performance of our proposed method on environments with kinematic and morphology shifts. We first introduce the experimental setup, including the environments and baselines. Then, we present the adaptation performance of our method compared to the baselines. A parameter study is also conducted to analyze the impact of different parameters on the performance of our method.

### 5.1 EXPERIMENTAL SETUP

We conduct experiments in four environments (*ant*, *hopper*, *halfcheetah*, *walker*) from Gym MuJoCo (Todorov et al., 2012; Brockman et al., 2016). The source domain is set as the original environment, while the target domain is set as the environment with kinematic or morphology shifts. The kinematic shift is achieved by limiting the rotation range of the joints, while the morphology shift is achieved by clipping the size of some limbs. We provide the setting details in Appendix E.1.

We compare our method with the following baselines: **DARC** (Eysenbach et al., 2021), which trains domain classifiers to estimate the dynamics discrepancy and modifies the reward function in the source domain; **VGDF** (Xu et al., 2023), which uses a value-guided data filtering method to select data from the source domain; **PAR** (Lyu et al., 2024a), which trains encoders to estimate the representation discrepancy and modifies the reward function in the source domain; **SAC-IW**, which estimates the dynamics discrepancy as an importance sampling term for value function; **SAC-tune**, which fine-tunes the policy in the target domain for $10^5$ environmental steps; **SAC-tar** (Haarnoja et al., 2018), which is the vanilla SAC trained in the target domain with $10^5$ environmental steps; **Oracle** (Haarnoja et al., 2018), which is the vanilla SAC trained in the target domain with 1M environmental steps. We implement all algorithms based on the official code of ODRL (Lyu et al., 2024c) and follow the hyperparameters in the original paper. We allow all algorithms to interact with the source domain for 1M environmental steps and the target domain for $10^5$ environmental steps, *i.e.*, the target domain interaction frequency $F = 10$. All algorithms are trained with five random seeds. Implementation details are provided in Appendix E.2.

### 5.2 ADAPTATION PERFORMANCE EVALUATION

We conduct experiments on eight tasks with kinematic and morphology shifts to evaluate the adaptation performance of DADiff and baselines. The results are presented in Figure 2. Notably, our proposed method demonstrates superior or highly competitive performance against all baselines in the majority of tasks. We further discuss the performance of two variants of DADiff, DADiff-modify and DADiff-select, respectively.

**Reward modification variant.** The reward modification variant of our method, DADiff-modify, consistently outperforms other reward modification baselines across all tasks and remains competitive with oracle methods. As illustrated in Figure 2, DADiff-modify shows particularly strong and consistent performance. It outperforms other reward modification methods, including PAR, DARC,

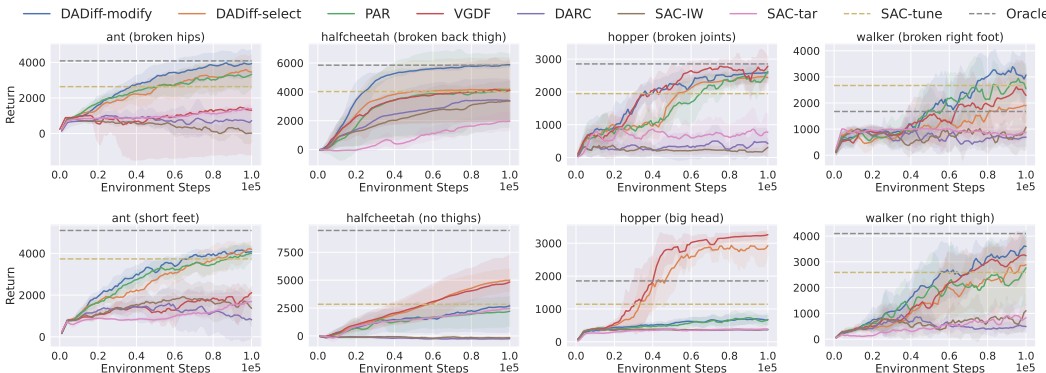

Figure 2: Adaptation performance on kinematic (top) and morphology (bottom) shifts. The solid curves and the shaded regions denote the mean and standard deviation over five random seeds, respectively. DADiff demonstrates superior or highly competitive performance against all baselines in the majority of tasks.

and SAC-IW, across all eight tasks. We also provide a quantitative improvement analysis for the specific cases of *ant(broken hips)* and *walker(broken right foot)*. Our improvements over PAR in these two settings are +637.54 (corresponding to a 19.1% improvement over PAR) and +447.1 (corresponding to a 15.2% improvement over PAR), respectively. When compared to oracle methods, DADiff-modify consistently surpasses SAC-tune and SAC-tar as well. To further explore the performance of DADiff-modify in stochastic environments, we provide an experiment in Section 6.

**Data selection variant.** In Figure 2, the data selection variant, DADiff-select, proves to be a highly effective alternative by achieving competitive performance against top baselines, especially in tasks where reward modification methods falter. Specifically, in the *halfcheetah (no thighs)* and *hopper (big head)* tasks, reward modification methods exhibit poor performance. In contrast, DADiff-select achieves results that are highly competitive with the top-performing baseline, VGDF. This suggests that in certain tasks, directly filtering for transitions with low dynamics mismatch is a more effective strategy than modifying rewards. Additionally, while the VGDF demonstrates top-tier performance in certain challenging tasks, specifically *hopper*

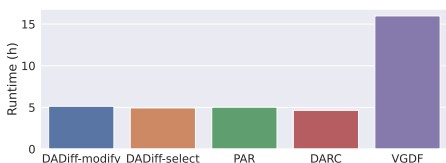

Figure 3: Runtime comparison on the *halfcheetah (broken back thigh)* task. VGDF requires $3\times$ more training time than other methods due to its model-based approach.

*(big head)* and *halfcheetah (no thighs)*, its approach carries significant trade-offs. Since VGDF is a model-based approach, it takes significantly longer to train by more than $3\times$, as shown in Figure 3. On the other hand, DADiff-select is able to match or exceed the performance of VGDF on such environments while maintaining comparable efficiency to similar model-free baselines. We further provide additional computational cost analysis in Appendix F.1.

### 5.3 PARAMETER STUDY

The performance of DADiff is influenced by several key hyperparameters. To better understand their roles, we conducted a series of experiments across different tasks. The results on *halfcheetah (broken back thigh)* and *walker (no right thigh)* are presented in Figure 4. More experimental results are provided in Appendix F.2.

**Penalty Coefficient** $\lambda$. $\lambda$ controls the scale of reward penalty in DADiff-modify. As shown in Figure 4a and Figure 9, Appendix F.2, we evaluate the performance of DADiff-modify across multiple values of $\lambda$. We find that a worse performance is often shown in the setting $\lambda = 0$, where no penalty is adopted for rewards. It demonstrates the necessity of reward modification. Meanwhile, the results also indicate that the optimal value of $\lambda$ is task-dependent, and there could be multiple values that

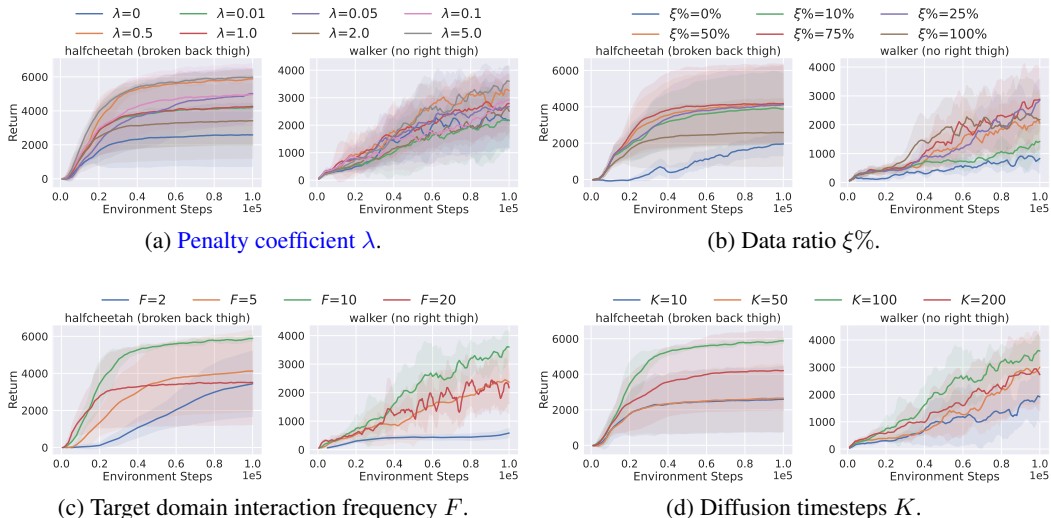

Figure 4: Parameter study. The solid curves and the shaded regions denote the mean and standard deviation over five random seeds, respectively.

yield good performance for a specific task. For instance, in the *halfcheetah (broken back thigh)* task, both $\lambda = 0.5$ and $\lambda = 5.0$ achieve the best performance. A poorly chosen $\lambda$ can significantly degrade performance, highlighting the importance of tuning this coefficient.

**Data Selection Ratio $\xi\%$.** $\xi\%$ controls the percentage of source domain data to retain in DADiff-select. As shown in Figure 4b and Figure 10, Appendix F.2, we evaluate the performance of DADiff-select across multiple values of $\xi\%$. Similar to the penalty coefficient, the optimal value of $\xi\%$ is task-dependent. We also find that both too much ($\xi\% = 100\%$) and too little (($\xi\% = 0\%$)) source data can lead to suboptimal performance. As retaining too much source data may introduce transitions with significant dynamics mismatch, while retaining too little may result in insufficient data for effective learning.

**Target Domain Interaction Frequency $F$.** $F$ controls how often policies interact with the target domain in both DADiff-modify and DADiff-select. Only $10^5$ interactions with the target domain are permitted and the interactions with the source domain are changed to adapt to different $F$. We provide the results of DADiff-modify in Figure 4c. We find that the frequency $F$ is best set to 10. This value provides the best performance, while collecting too much source domain data between target interactions ($F = 20$) can be detrimental, possibly causing the policy to diverge. Additional results of DADiff-select are provided in Figure 11a, Appendix F.2.

**Diffusion Timesteps $K$.** $K$ controls the number of diffusion timesteps used to measure the discrepancy in both DADiff-modify and DADiff-select. We provide the results of DADiff-modify in Figure 4d. The results shows that performance improves up to $K = 100$. Increasing $K$ further to 200 causes a decline, likely due to the limited capacity of the noise model, which may struggle to accurately estimate noise across too many timesteps. Additional results of DADiff-select are provided in Figure 11b, Appendix F.2.

## 6 DISCUSSIONS

**Connection between DADiff and PAR.** We explore the connection between PAR and our method from a theoretical perspective. The performance bound of our method is controlled by the generative trajectory discrepancy in Theorem 4.2. We consider a special case, where the number of latent states in the trajectory is $K = 1$. Instead of considering latent states in the generative trajectory, we take $s'_1$ as a latent representation and introduce the one-to-one representation mapping assumption in PAR (Lyu et al., 2024a), which assumes that there exists a one-to-one mapping for each state-action pair

Table 1: We report the adaptation performance with stochastic dynamics controlled by the standard deviation parameter $\varsigma$. The average return and standard deviation over five random seeds are reported. The best results are highlighted in **bold**. The relative performance change compared to the deterministic setting ($\varsigma = 0.0$) is reported in parentheses. For both environments, DADiff-modify shows a smaller decrease in performance than PAR.

| | (a) hopper (broken joints) | | | (b) walker (broken right foot) | |
|---|---|---|---|---|---|
| $\varsigma$ | **DADiff-modify** | **PAR** | $\varsigma$ | **DADiff-modify** | **PAR** |
| 0.00 | 2582.1±251.6 | **2623.1**±105.2 | 0.00 | **3390.4**±464.4 | 2943.3±546.7 |
| 0.01 | **2591.0**±159.2 (↑ 0.34%) | 2398.3±297.8 (↓ 8.57%) | 0.01 | **2879.3**±688.9 (↓ 15.08%) | 2373.8±1072.4 (↓ 19.35%) |
| 0.02 | **2515.9**±101.8 (↓ 2.57%) | 2328.7±302.9 (↓ 11.22%) | 0.02 | 2812.5±934.6 (↓ 17.05%) | **2825.8**±466.6 (↓ 3.99%) |
| 0.03 | **2574.2**±280.6 (↓ 0.31%) | 2406.1±455.7 (↓ 8.27%) | 0.03 | **3176.8**±796.4 (↓ 6.30%) | 1613.9±878.7 (↓ 45.17%) |

$(s, a)$ and its latent representation $s'_1$. In this setting, the state-action pair $(s, a)$ in Equation 5 can be all replaced by the corresponding latent representation $s'_1$. Therefore, the performance bound can be rewritten as follows,

$$\eta_{\mathcal{M}_{\text{src}}}(\pi) - \eta_{\mathcal{M}_{\text{tar}}}(\pi) \leq \frac{\sqrt{2}\gamma r_{\max}}{(1-\gamma)^2} \mathbb{E}_{\rho^\pi_{\text{src}}} \left[ \sqrt{\mathbb{E}_{P_{\text{src}}} \left[ D_{\text{KL}}(P_{\text{src}}(s'_0|s'_1) || P_{\text{tar}}(s'_0|s'_1)) \right]} \right]. \quad (15)$$

We further introduce a conclusion proven in PAR (Lyu et al., 2024a), which is formulated as follows,

$$D_{\text{KL}}(P_{\text{src}}(s'_1|s'_0) || P_{\text{tar}}(s'_1|s'_0)) = D_{\text{KL}}(P_{\text{src}}(s'_0|s'_1) || P_{\text{tar}}(s'_0|s'_1)) + \mathbb{H}(s'_{\text{src}}) - \mathbb{H}(s'_{\text{tar}}). \quad (16)$$

Therefore, the performance bound can be rewritten as follows,

$$\begin{aligned}
\eta_{\mathcal{M}_{\text{src}}}(\pi) - \eta_{\mathcal{M}_{\text{tar}}}(\pi) &\leq \frac{\sqrt{2}\gamma r_{\max}}{(1-\gamma)^2} \mathbb{E}_{\rho^\pi_{\text{src}}} \left[ \sqrt{\mathbb{E}_{P_{\text{src}}} \left[ D_{\text{KL}}(P_{\text{src}}(s'_1|s'_0) || P_{\text{tar}}(s'_1|s'_0)) \right]} \right] \\
&+ \frac{\sqrt{2}\gamma r_{\max}}{(1-\gamma)^2} \mathbb{E}_{\rho^\pi_{\text{src}}} \left[ \sqrt{\mathbb{E}_{P_{\text{src}}} \left[ \mathbb{H}(s'_{\text{src}}) - \mathbb{H}(s'_{\text{tar}}) \right]} \right].
\end{aligned} \quad (17)$$

This performance bound is consistent with the performance bound of PAR, which indicates that PAR can be considered as a special case of our method. However, the one-to-one representation mapping assumption may not hold in practice, especially in stochastic environments, which limits the application of PAR. In contrast, our method does not rely on this assumption and can handle more general scenarios. We validate this point in environments with stochastic dynamics. Noises with different standard deviation $\varsigma$ are introduced to the actions to simulate stochastic dynamics, and two tasks with kinematic shifts, *hopper (broken joints)* and *walker (broken right foot)*, are considered. We evaluate the performance of DADiff-modify and PAR, which is presented in Table 1. Notably, our method maintains robust performance even as the standard deviation $\varsigma$ increases, while PAR's performance degrades significantly. We believe the decrease in PAR's performance is due to its reliance on one-to-one representation assumptions, which may not hold in stochastic settings. We provide more results on stochastic dynamics in Appendix F.3.

**Reward distribution analysis.** We further examine the reasons behind the superior performance of DADiff-select, in contrast to the severe failure of DADiff-modify on *halfcheetah (no thighs)* and *hopper (big head)* tasks, as illustrated in Figure 2. Specifically, we analyze the reward distributions of source-domain data after modification or selection. The results are presented in Figure 5. We find that DADiff-select generates a higher distribution in the low-reward region compared to DADiff-modify on both tasks. This suggests that the low-reward data may play a crucial role in these tasks, which can ef-

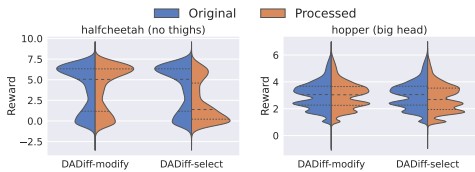

Figure 5: Reward distribution comparison between the source-domain rewards before processing (Original) and after modification or selection (Processed).

fectively guide the policy to avoid undesirable states and actions. More results on reward distribution are provided in Appendix F.4.

## 7    CONCLUSION

This work explores the problem of online dynamics adaptation in reinforcement learning from a generative modeling perspective. We first theoretically analyze the performance bound of a policy in the source and target domains, which is controlled by the generative trajectory discrepancy. Based on this analysis, we propose a novel method, DADiff, which utilizes diffusion models to measure the dynamics discrepancy and performs either reward modification or data selection to adapt to the target domain. Extensive experiments demonstrate that our method outperforms existing baselines in various tasks with kinematic and morphology shifts. We also conduct a parameter study and multiple discussions to further explore the properties of our method.

## ETHICS STATEMENT

This research focuses on an online dynamics adaptation problem in reinforcement learning, which is a fundamental problem in the field of sim-to-real transfer. We believe that our work can contribute to the development of more robust and adaptable reinforcement learning algorithms, which can be beneficial for various applications. However, we also acknowledge that the deployment of reinforcement learning algorithms in real-world environments may raise ethical concerns, such as safety and fairness. We encourage researchers and practitioners to consider these ethical implications when applying our method in practice.

## REPRODUCIBILITY STATEMENT

Our code, data, and instructions needed to reproduce the main experimental results are included in the supplementary material. We provide detailed descriptions of the algorithms, experimental setup, and hyperparameters in the main text and appendix. Proofs of the theoretical results are also provided in the appendix to ensure the reproducibility of our work.

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

## A    THE USE OF LARGE LANGUAGE MODELS

Large Language Models (LLMs) were utilized in the preparation of this manuscript. Specifically, LLMs were employed to assist in refining the clarity and coherence of the text, ensuring that complex ideas were communicated effectively. The use of LLMs was limited to language editing and did not influence the scientific content or conclusions of the work. All technical details, experimental results, and theoretical analyses were developed independently by the authors. We acknowledge the assistance of LLMs in enhancing the readability of the manuscript while maintaining the integrity of the research presented.

## B    PROOFS OF THE PERFORMANCE GUARANTEES

In this section, we provide detailed proofs of the performance guarantees stated in the main text. For readability, we restate theorems and provide some lemmas that are used in the proofs.

### B.1    USEFUL LEMMAS

**Lemma B.1 (Telescoping lemma.)** Denote $\mathcal{M}_1 = (\mathcal{S}, \mathcal{A}, P_1, r, \gamma)$ and $\mathcal{M}_2 = (\mathcal{S}, \mathcal{A}, P_2, r, \gamma)$ as two MDPs with the same state and action spaces but different transition dynamics $P_1$ and $P_2$. The performance difference of a policy $\pi$ evaluated in $\mathcal{M}_1$ and $\mathcal{M}_2$ can be expressed as:

$$\eta_{\mathcal{M}_1}(\pi) - \eta_{\mathcal{M}_2}(\pi) = \frac{\gamma}{1-\gamma} \mathbb{E}_{\rho^\pi_{\mathcal{M}_1}(s,a)} \left[ \mathbb{E}_{s' \sim P_1}[V^\pi_{\mathcal{M}_2}(s')] - \mathbb{E}_{s' \sim P_2}[V^\pi_{\mathcal{M}_2}(s')] \right]$$

*Proof.* Please see Lemma 4.3 in SLBO (Luo et al., 2019) for a detailed proof.

**Lemma B.2 (KL divergence of Gaussian distributions.)** Specify two normal distributions $P_a = \mathcal{N}(\mu_a, \Sigma)$ and $P_b = \mathcal{N}(\mu_b, \Sigma)$, which have different means $\mu_a$ and $\mu_b$ but share the same covariance matrix $\Sigma = \sigma^2 I$ with $\sigma^2$ being a predefined scalar. The KL divergence between $P_a$ and $P_b$ can be written as below,

$$D_{\mathrm{KL}}(P_a || P_b) = \frac{1}{2\sigma^2} \| \mu_a - \mu_b \|_2^2.$$

*Proof.* According to the definition of KL divergence between two multivariate Gaussian distributions, we have

$$D_{\mathrm{KL}}(P_a || P_b) = \frac{1}{2} \left( \log \frac{|\Sigma_b|}{|\Sigma_a|} - \mathrm{tr}(I) + \mathrm{tr}(\Sigma_b^{-1} \Sigma_a) + (\mu_b - \mu_a)^\top \Sigma_b^{-1} (\mu_b - \mu_a) \right)$$

$$= \frac{1}{2} \left( \frac{1}{\sigma^2} \| \mu_a - \mu_b \|_2^2 \right)$$

$$= \frac{1}{2\sigma^2} \| \mu_a - \mu_b \|_2^2.$$

### B.2    PROOF OF THEOREM 4.2

**Theorem B.3 (Performance bound controlled by generative trajectory discrepancy.)** *Denote $\mathcal{M}_{\mathrm{src}}$ and $\mathcal{M}_{\mathrm{tar}}$ as the source and target domains with different dynamics, respectively. The performance difference of any policy $\pi$ evaluated in $\mathcal{M}_{\mathrm{src}}$ and $\mathcal{M}_{\mathrm{tar}}$ can be bounded as below,*

$$\eta_{\mathcal{M}_{\mathrm{src}}}(\pi) - \eta_{\mathcal{M}_{\mathrm{tar}}}(\pi) \le \frac{\sqrt{2}\gamma r_{\max}}{(1-\gamma)^2} \underbrace{\mathbb{E}_{\rho^\pi_{\mathrm{src}}} \left[ \sqrt{\mathbb{E}_{P_{\mathrm{src}}} \left[ D_{\mathrm{KL}}(P_{\mathrm{src}}(s'_K|s,a) || P_{\mathrm{tar}}(s'_K|s,a)) \right]} \right]}_{(a):\ \text{initial latent state deviation}}$$

$$+ \frac{\sqrt{2}\gamma r_{\max}}{(1-\gamma)^2} \underbrace{\mathbb{E}_{\rho^\pi_{\mathrm{src}}} \left[ \sqrt{\mathbb{E}_{P_{\mathrm{src}}} \left[ \sum_{k=1}^{K} D_{\mathrm{KL}}(P_{\mathrm{src}}(s'_{k-1}|s'_k, s, a) || P_{\mathrm{tar}}(s'_{k-1}|s'_k, s, a)) \right]} \right]}_{(b):\ \text{latent state transition mismatch}}$$

**Proof.** As the value function $V_{\mathcal{M}}^{\pi}(s)$ estimates the expected return of a policy $\pi$ starting from state $s$ in domain $\mathcal{M}$, and the rewards are bounded, we have $|V_{\mathcal{M}}^{\pi}(s)| \leq r_{\max}/(1-\gamma), \forall s$. By using Lemma B.1, we have:

$$\eta_{\mathcal{M}_{\text{src}}}(\pi) - \eta_{\mathcal{M}_{\text{tar}}}(\pi) = \frac{\gamma}{1-\gamma}\mathbb{E}_{\rho_{\text{src}}^{\pi}}\left[\mathbb{E}_{P_{\text{src}}}[r(s,a)] - \mathbb{E}_{P_{\text{tar}}}[r(s,a)]\right]$$

$$= \frac{\gamma}{1-\gamma}\mathbb{E}_{\rho_{\text{src}}^{\pi}}\left[\int_{s_0'}P_{\text{src}}(s_0'|s,a)V_{\text{tar}}^{\pi}(s_0') - \int_{s_0'}P_{\text{tar}}(s_0'|s,a)V_{\text{tar}}^{\pi}(s_0')ds_0'\right]$$

$$\leq \frac{\gamma}{1-\gamma}\mathbb{E}_{\rho_{\text{src}}^{\pi}}\left[\int_{s_0'}\left(P_{\text{src}}(s_0'|s,a) - P_{\text{tar}}(s_0'|s,a)\right)|V_{\text{tar}}^{\pi}(s_0')|\,ds_0'\right]$$

$$\leq \frac{\gamma r_{\max}}{(1-\gamma)^2}\mathbb{E}_{\rho_{\text{src}}^{\pi}}\left[\int_{s_0'}P_{\text{src}}(s_0'|s,a) - P_{\text{tar}}(s_0'|s,a)ds_0'\right]$$

$$= \frac{\gamma r_{\max}}{(1-\gamma)^2}\mathbb{E}_{\rho_{\text{src}}^{\pi}}\left[\int_{s_{0:K}'}P_{\text{src}}(s_{0:K}'|s,a) - P_{\text{tar}}(s_{0:K}'|s,a)ds_{0:K}'\right]$$

$$= \frac{2\gamma r_{\max}}{(1-\gamma)^2}\mathbb{E}_{\rho_{\text{src}}^{\pi}}\left[D_{\text{TV}}(P_{\text{src}}(s_{0:K}'|s,a)||P_{\text{tar}}(s_{0:K}'|s,a))\right]$$

$$\leq \frac{\sqrt{2}\gamma r_{\max}}{(1-\gamma)^2}\mathbb{E}_{\rho_{\text{src}}^{\pi}}\left[\sqrt{D_{\text{KL}}(P_{\text{src}}(s_{0:K}'|s,a)||P_{\text{tar}}(s_{0:K}'|s,a))}\right] \quad\text{(a)}$$

$$= \frac{\sqrt{2}\gamma r_{\max}}{(1-\gamma)^2}\mathbb{E}_{\rho_{\text{src}}^{\pi}}\left[\sqrt{\mathbb{E}_{P_{\text{src}}}\left[\log\frac{P_{\text{src}}(s_{0:K}'|s,a)}{P_{\text{tar}}(s_{0:K}'|s,a)}\right]}\right]$$

$$= \frac{\sqrt{2}\gamma r_{\max}}{(1-\gamma)^2}\mathbb{E}_{\rho_{\text{src}}^{\pi}}\left[\sqrt{\mathbb{E}_{P_{\text{src}}}\left[\log\frac{P_{\text{src}}(s_K'|s,a)}{P_{\text{tar}}(s_K'|s,a)} + \sum_{k=1}^{K}\log\frac{P_{\text{src}}(s_{k-1}'|s_k',s,a)}{P_{\text{tar}}(s_{k-1}'|s_k',s,a)}\right]}\right] \quad\text{(b)}$$

$$\leq \frac{\sqrt{2}\gamma r_{\max}}{(1-\gamma)^2}\mathbb{E}_{\rho_{\text{src}}^{\pi}}\left[\sqrt{\mathbb{E}_{P_{\text{src}}}[D_{\text{KL}}(P_{\text{src}}(s_K'|s,a)||P_{\text{tar}}(s_K'|s,a))]}\right]$$

$$+ \frac{\sqrt{2}\gamma r_{\max}}{(1-\gamma)^2}\mathbb{E}_{\rho_{\text{src}}^{\pi}}\left[\sqrt{\mathbb{E}_{P_{\text{src}}}\left[\sum_{k=1}^{K}D_{\text{KL}}(P_{\text{src}}(s_{k-1}'|s_k',s,a)||P_{\text{tar}}(s_{k-1}'|s_k',s,a))\right]}\right] \quad\text{(c)}$$

where $D_{\text{TV}}(P||Q)$ is the total variation distance between two distributions $P$ and $Q$, the step (a) holds by Pinsker's inequality (Csiszár & Körner, 2011), the step (b) holds by the Markov property, and the step (c) holds by the subadditivity of the square root function. The proof shows that the performance difference can be controlled by the distributional divergence of latent states in generative trajectories.

## C  EXTENDED IMPLEMENTATION

**Flow Matching**  We further extend Theorem 4.2 to continuous-time generative models, *i.e.*, flow matching (Lipman et al., 2022; Liu et al., 2023), and provide the implementation details in this section. To better clarification, we redefine the timestep $k \in [0,1]$ in flow matching, which is different from the discrete timestep $k \in \{0, 1, 2, \ldots, K\}$ in diffusion models. Flow matching model learns a vector field to transform a standard Gaussian distribution to a complex distribution. Specifically, given a data point $x_0$, flow matching constructs a continuous-time flow from a standard Gaussian distribution to the data point $x_0$, which is defined as follows,

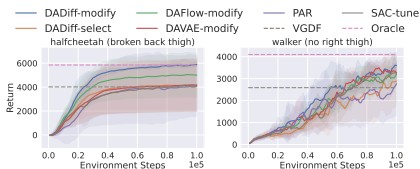

Figure 6: Adaptation performance of DAFlow-modify and DAVAE-modify. The solid curves and the shaded regions denote the mean and standard deviation over five random seeds, respectively.

$$x_{k-\Delta k} = x_k + \Delta k \cdot v_\theta(x_k, k),$$

where $\Delta k$ is a small step size, $x_k = (1-k)x_0 + kx_1$, $x_1 \sim \mathcal{N}(0, I)$, and $v_\theta(x_k, k)$ is the vector field that is estimated by a neural network and indicates the direction of the flow at point $x_k$. The flow matching model is trained to minimize the following objective,

$$\mathcal{L}_{\text{flow}} = \mathbb{E}_{x_0, k, \epsilon} \left[ \|(x_1 - x_0) - v_\theta(x_k, k)\|^2 \right].$$

We follow the same procedure in Section 4.2 to measure the dynamics discrepancy based on the trained flow matching model. Specifically, the vector field in the source domain is denoted as $v_{\text{src}}(s'_k, k)$, which is constructed by the source domain data, *i.e.*, $v_{\text{src}}(s'_k, k) = \mathbb{E}[s'_1 - s'_0 | s'_k]$, where $s'_0$ is the next state in the source domain, and $s'_1 \sim \mathcal{N}(0, I)$. Meanwhile, the vector field in the target domain is estimated by a neural network $v^\theta_{\text{tar}}(s'_k, k)$, which is trained on the target domain data. The generative trajectory deviation $d(s, a, s')$ is formulated as follows,

$$d(s, a, s') = \mathbb{E}_{s'_1} \left[ \sum_{k \in \{\Delta k, 2\Delta k, \ldots, 1\}} \Delta k^2 \|(s'_1 - s'_0) - v^\theta_{\text{tar}}((1-k)s'_0 + ks'_1, k)\|^2 \right].$$

Similarly, the deviation $d(s, a, s')$ can be used for reward modification and data selection in the same way as in Section 4.2. We provide the domain adaptation performance of the reward modification variant based on flow matching (DAFlow-modify) in Figure 6. The results show that DAFlow-modify achieves better performance compared to the baseline methods, which demonstrates the generality of our theoretical findings. In addition, DAFlow-modify has slightly inferior performance compared to DADiff-modify, showing that diffusion models may be more effective in measuring the dynamics discrepancy for domain adaptation in reinforcement learning.

**Conditional VAE** To further investigate the capability of deviation measurement across different generative models, we also provide an implementation based on conditional VAE (Kingma & Welling, 2013). Consistently, we adopt the standard Gaussian distribution as the initial latent state, *i.e.*, $s'_1 \sim \mathcal{N}(0, I)$. We follow the same procedure in Section 4.2 to measure the dynamics discrepancy based on the trained decoder $p^\theta_{\text{tar}}(s'_0 | s'_1, s, a)$ of conditional VAE. The deviation $d(s, a, s')$ can be fomulated as,

$$d(s, a, s') = \mathbb{E}_{s'_1} \left[ \|s'_0 - p^\theta_{\text{tar}}(s'_0 | s'_1, s, a)\|^2 \right].$$

We report the results of the reward modification variant based on conditional VAE (DAVAE-modify) in Figure 6 as well. The results demonstrate that DADiff-modify achieves the best performance across different implementations, which consistently indicates that diffusion models are the better choice to model the generative trajectory deviation.

## D PSEUDOCODES

In this section, we provide the pseudocode of our proposed method in Algorithm 1, including both reward modification and data selection variants.

## E EXPERIMENTAL DETAILS

In this section, we provide detailed experimental settings, including environment settings, implementation details, and hyperparameter settings.

### E.1 ENVIRONMENT SETTING

Four environments from OpenAI Gym (Brockman et al., 2016) are considered in our experiments, including *ant-v3*, *halfcheetah-v2*, *hopper-v2* and *walker2d-v2*, which are simulated by the MuJoCo physics engine (Todorov et al., 2012). These environments are also widely used in previous works on domain adaptation in reinforcement learning (Xu et al., 2023; Lyu et al., 2024a). To evaluate the effectiveness of our proposed method under different dynamics shifts, we adopt the original environments as the source domain, and both kinematic shifts and morphology shifts are considered to construct the target domain. Specifically, the kinematic shifts are introduced by modifying the joint rotation angles of the robots, while the morphology shifts are introduced by changing the sizes of some body parts. The details of the target domain settings are summarized as follows:

---

**Algorithm 1** Domain Adaptation with Diffusion (DADiff)

---

**Input:** Source domain $\mathcal{M}_{\text{src}}$, target domain $\mathcal{M}_{\text{tar}}$, and target domain interaction frequency $F$
**Initialization:** Policy $\pi$, value function $\{Q_{\phi_i}\}_{i=1,2}$, target value function $\{Q_{\phi'_i}\}_{i=1,2}$, noise model $\epsilon_\theta$, replay buffers $\{\mathcal{D}_{\text{src}}, \mathcal{D}_{\text{tar}}\}$, data selection ratio $\xi$, batch size $N$
1: **for** $i = 1, 2, \ldots$ **do**
2:     Collect $(s_{\text{src}}, a_{\text{src}}, r_{\text{src}}, s'_{\text{src}})$ from the source domain $\mathcal{M}_{\text{src}}$ and store in $\mathcal{D}_{\text{src}}$
3:     **if** $i \mod F = 0$ **then**
4:         Collect $(s_{\text{tar}}, a_{\text{tar}}, r_{\text{tar}}, s'_{\text{tar}})$ from the target domain $\mathcal{M}_{\text{tar}}$ and store in $\mathcal{D}_{\text{tar}}$
5:     **end if**
6:     Sample $N$ transitions from $\mathcal{D}_{\text{tar}}$ to train the noise model $\epsilon_{\text{tar}}^\theta$ via Equation 9
7:     Sample $N$ transitions from $\mathcal{D}_{\text{src}}$ to obtain the deviation $d(s_{\text{src}}, a_{\text{src}}, s'_{\text{src}})$ via Equation 10
8:     **if** using reward modification **then**           ▷ reward modification
9:         Modify source domain rewards via Equation 11
10:        Update value functions $Q_{\phi_i}$ by minimizing Equation 12
11:     **else**                           ▷ data selection
12:         Select the $\xi$-quantile data from the source domain based on $d(s_{\text{src}}, a_{\text{src}}, s'_{\text{src}})$
13:         Update value functions $Q_{\phi_i}$ by minimizing Equation 13
14:     **end if**
15:     Update actor $\pi$ by minimizing Equation 14
16:     Update target value functions $Q_{\phi'_i}$
17: **end for**

---

- ***ant (broken hips)***: We modify the joint rotation angles of the hips on two legs from $[-30, 30]$ to $[-0.3, 0.3]$.

- ***halfcheetah (broken back thigh)***: We modify the joint rotation angle of the back thigh from $[-0.52, 1.05]$ to $[-0.0052, 0.0105]$.

- ***hopper (broken joints)***: We modify the joint rotation angles of the head and foot from $[-150, 0], [-45, 45]$ to $[-0.15, 0], [-18, 18]$, respectively.

- ***walker (broken right foot)***: We modify the joint rotation angle of the foot on the right leg from $[-45, 45]$ to $[-0.45, 0.45]$.

- ***ant (short feet)***: We modify the sizes of the feet on the front two legs of the robot, which are shown below:

```
<!-- leg 1 -->
<geom fromto="0.0 0.0 0.0 0.1 0.1 0.0" name="left_ankle_geom" size="0.08"
    type="capsule"/>
<!-- leg 2 -->
<geom fromto="0.0 0.0 0.0 -0.1 0.1 0.0" name="right_ankle_geom" size="
    0.08" type="capsule"/>
```

- ***halfcheetah (no thighs)***: We modify the sizes of the back thigh and the forward thigh of the robot, which are shown below:

```
<!-- back thigh -->
<geom fromto="0 0 0 -0.0001 0 -0.0001" name="bthigh" size="0.046" type="
    capsule"/>
<body name="bshin" pos="-0.0001 0 -0.0001">
<!-- forward thigh -->
<geom fromto="0 0 0 0.0001 0 0.0001" name="fthigh" size="0.046" type="
    capsule"/>
<body name="fshin" pos="0.0001 0 0.0001">
```

- ***hopper (big head)***: We modify the size of the head of the robot, which is shown below:

```
<!-- head size -->
<geom friction="0.9" fromto="0 0 1.45 0 0 1.05" name="torso_geom" size="
    0.125" type="capsule"/>
```

- *walker (no right foot)*: We modify the size of the thigh on the right leg of the robot, which is shown below:

```
<!-- right leg -->
<body name="thigh" pos="0 0 1.05">
 <joint axis="0 -1 0" name="thigh_joint" pos="0 0 1.05" range="-150 0"
     type="hinge"/>
 <geom friction="0.9" fromto="0 0 1.05 0 0 1.045" name="thigh_geom" size=
     "0.05" type="capsule"/>
 <body name="leg" pos="0 0 0.35">
  <joint axis="0 -1 0" name="leg_joint" pos="0 0 1.045" range="-150 0"
      type="hinge"/>
  <geom friction="0.9" fromto="0 0 1.045 0 0 0.3" name="leg_geom" size="
      0.04" type="capsule"/>
  <body name="foot" pos="0.2 0 0">
   <joint axis="0 -1 0" name="foot_joint" pos="0 0 0.3" range="-45 45"
       type="hinge"/>
   <geom friction="0.9" fromto="-0.0 0 0.3 0.2 0 0.3" name="foot_geom"
       size="0.06" type="capsule"/>
  </body>
 </body>
</body>
```

Detailed modifications of the `xml` files for the target domains are provided in the supplementary material.

## E.2 IMPLEMENTATION DETAILS

In this section, we provide the implementation details of our proposed method and baselines. All methods are implemented based on the Soft Actor-Critic (SAC) algorithm (Haarnoja et al., 2018), which is a widely used off-policy reinforcement learning algorithm. The details are summarized as follows:

- **PAR:** PAR is constructed based on the theoretical analysis that the performance difference between source and target domains can be bounded by the representation discrepancy, *i.e.*,

$$\eta_{\mathcal{M}_{\mathrm{src}}}(\pi) - \eta_{\mathcal{M}_{\mathrm{tar}}}(\pi) \leq \frac{\sqrt{2}\gamma r_{\max}}{(1-\gamma)^2} \mathbb{E}_{\rho_{\mathrm{src}}^\pi} \left[ \sqrt{\mathbb{E}_P \left[ D_{\mathrm{KL}}(P(z|s'_{\mathrm{src}})||P(z|s'_{\mathrm{tar}})) \right]} \right]$$
$$+ \frac{\sqrt{2}\gamma r_{\max}}{(1-\gamma)^2} \mathbb{E}_{\rho_{\mathrm{src}}^\pi} \left[ \sqrt{\mathbb{H}(s'_{\mathrm{src}}) + \mathbb{H}(s'_{\mathrm{tar}})} \right],$$

where $z'$ is the latent representation of the state-action pair, $s'_{\mathrm{src}}$ and $s'_{\mathrm{tar}}$ are the next states in source and target domains, respectively, and $\mathbb{H}(\cdot)$ is the entropy. PAR learns a shared state encoder $f_\phi$ and a state-action encoder $g_\theta$ to obtain the latent representations of states and state-action pairs, respectively. The encoders are trained to minimize the representation discrepancy in the target domain. The source domain rewards are modified by adopting a reward penalty via:

$$r_{\mathrm{mod}}(s, a, s'_{\mathrm{src}}) = r(s, a, s'_{\mathrm{src}}) - \lambda \cdot \left[ f_\phi(g_\theta(s'_{\mathrm{src}}), a_{\mathrm{src}}) - g_\theta(s'_{\mathrm{tar}}) \right]^2,$$

where $\lambda$ is a hyperparameter to balance the original reward and the penalty. We use the official code of ODRL (Lyu et al., 2024c) to implement PAR, and we follow the default hyperparameter settings provided in PAR.

- **VGDF:** VGDF is constructed based on the theoretical analysis that the performance difference between source and target domains can be bounded by the value discrepancy, *i.e.*,

$$\eta_{\mathcal{M}_{\mathrm{src}}}(\pi) - \eta_{\mathcal{M}_{\mathrm{tar}}}(\pi) \leq \frac{\gamma}{1-\gamma} \mathbb{E}_{\rho_{\mathrm{src}}^\pi} \left[ |\mathbb{E}_{P_{\mathrm{src}}} [V_{\mathrm{src}}^\pi(s')] - \mathbb{E}_{P_{\mathrm{src}}} [V_{\mathrm{tar}}^\pi(s')] | \right].$$

VGDF learns an ensemble of probabilistic dynamics models to predict the next state in the target domain, which is used to estimate the value discrepancy, and selects source domain data with small value discrepancy to train the critic via:

$$\mathcal{L}_{\mathrm{critic}} = \mathbb{E}_{(s,a,r,s')\sim\mathcal{D}_{\mathrm{tar}}} \left[ (Q_\phi - \mathcal{T}Q_\phi) \right] + \mathbb{E}_{(s,a,r,s')\sim\mathcal{D}_{\mathrm{src}}} \left[ \omega(s,a,s')(Q_\phi - \mathcal{T}Q_\phi) \right],$$

where $\omega(s, a, s') = \mathbb{1}(\Lambda(s, a, s') \leq \Lambda_{\xi\%})$ is the data selection function, $\Lambda(s, a, s')$ is the value discrepancy estimated by the learned dynamics models, $\xi\%$ is the data selection ratio, $\mathcal{T}$ is the Bellman operator, and $\mathcal{D}_{\text{tar}}$ and $\mathcal{D}_{\text{src}}$ are the replay buffers of target and source domains, respectively. We use the official code of ODRL (Lyu et al., 2024c) to implement VGDF, and we follow the default hyperparameter settings provided in VGDF to set the data selection ratio as 25%.

• **DARC:** DARC estimates the reward correction term via two domain classifiers $q_{\theta_{\text{SA}}}(\cdot|s, a)$ and $q_{\theta_{\text{SAS}}}(\cdot|s, a, s')$, which are trained to distinguish the source and target domain data. These two classifiers are trained via:

$$\mathcal{L}_{\text{SA}} = -\mathbb{E}_{(s,a)\sim\mathcal{D}_{\text{tar}}}\left[\log q_{\theta_{\text{SA}}}(\text{target}|s, a)\right] - \mathbb{E}_{(s,a)\sim\mathcal{D}_{\text{src}}}\left[\log q_{\theta_{\text{SA}}}(\text{source}|s, a)\right],$$

$$\mathcal{L}_{\text{SAS}} = -\mathbb{E}_{(s,a,s')\sim\mathcal{D}_{\text{tar}}}\left[\log q_{\theta_{\text{SAS}}}(\text{target}|s, a, s')\right] - \mathbb{E}_{(s,a,s')\sim\mathcal{D}_{\text{src}}}\left[\log q_{\theta_{\text{SAS}}}(\text{source}|s, a, s')\right].$$

The source domain rewards are modified by adopting a reward correction via:

$$r_{\text{mod}}(s, a, s') = r(s, a, s') - \lambda \cdot \log \frac{q_{\theta_{\text{SA}}}(\text{source}|s, a)q_{\theta_{\text{SAS}}}(\text{target}|s, a, s')}{q_{\theta_{\text{SA}}}(\text{target}|s, a)q_{\theta_{\text{SAS}}}(\text{source}|s, a, s')}.$$

We use the official code of ODRL (Lyu et al., 2024c) to implement DARC, and we follow the default hyperparameter settings provided in PAR.

• **SAC-IW:** Different from DARC, SAC-IW estimates the importance weights via two domain classifiers $q_{\theta_{\text{SA}}}(\cdot|s, a)$ and $q_{\theta_{\text{SAS}}}(\cdot|s, a, s')$, which are trained to distinguish the source and target domain data. These two classifiers are trained via the same loss functions as DARC. The importance weights are estimated via:

$$\omega(s, a, s') = \frac{q_{\theta_{\text{SA}}}(\text{source}|s, a)q_{\theta_{\text{SAS}}}(\text{target}|s, a, s')}{q_{\theta_{\text{SA}}}(\text{target}|s, a)q_{\theta_{\text{SAS}}}(\text{source}|s, a, s')}.$$

The importance weights are used to reweight the error in the critic update via:

$$\mathcal{L}_{\text{critic}} = \mathbb{E}_{(s,a,r,s')\sim\mathcal{D}_{\text{src}}}\left[\omega(s, a, s')(Q_\phi - \mathcal{T}Q_\phi)\right].$$

To ensure the stablity of training, we clip the weights to the range $[1e^{-4}, 10]$. We use the official code of ODRL (Lyu et al., 2024c) to implement SAC-IW, and we follow the default hyperparameter settings provided in ODRL.

• **SAC-tar:** SAC-tar directly applies the SAC algorithm to interact with the target domain for $10^5$ environmental steps, without using any source domain data.

• **SAC-tune:** SAC-tune first pretrains the policy using the SAC algorithm in the source domain for 1M environmental steps, and then fine-tunes the pretrained policy in the target domain for another $10^5$ environmental steps.

• **DADiff-modify:** Instead of measuring the performance difference between the source and target domains via the representation discrepancy or value discrepancy, DADiff-modify measures it via the generative trajectory discrepancy in Equation 5. The noisy data points generated by the noise model are regarded as the latent states in our implementation. The noise model is trained to fit the target domain data via Equation 9. The source domain rewards are then modified by adopting a reward penalty via Equation 11. The value function is updated via Equation 12. We believe that the penalty coefficient $\lambda$ in Equation 11 is an important hyperparameter, and it is task-dependent. We conduct a hyperparameter search for $\lambda$ in $\{0.01, 0.05, 0.1, 0.5, 1.0, 2.0, 5.0\}$ for each task and report the adopted $\lambda$ in Table 2.

• **DADiff-select:** DADiff-select measures the performance difference between the source and target domains in the same way as DADiff-modify. The source domain data are selected based on the deviation and then used to train the value function, which is shown in Equation 13. As our method is more efficient in selecting source domain data, we conduct a hyperparameter search for the data selection ratio $\xi\%$ in $\{25\%, 50\%, 75\%\}$ for each task and report the adopted $\xi\%$ in Table 2.

### E.3 HYPERPARAMETER SETTINGS

The hyperparameter settings of our proposed method are summarized in Table 3. The hyperparameters of the baseline methods are set according to their original papers. For a fair comparison, we use the same hyperparameters for the SAC algorithm across all methods. In addition, we provide the adopted key hyperparameters for both reward modification and data selection variants of our proposed method in Table 2.

Table 2: Key hyperparameters of DADiff.

| Task Name | DADiff-select ($\xi\%$) | DADiff-modify ($\lambda$) | PAR ($\lambda$) | DARC ($\lambda$) |
|---|---|---|---|---|
| ant (broken hips) | 75% | 0.01 | 0.05 | 1.0 |
| ant (short feet) | 75% | 2.0 | 0.05 | 0.1 |
| halfcheetah (broken back thigh) | 75% | 0.5 | 5.0 | 2.0 |
| halfcheetah (no thighs) | 25% | 0.1 | 5.0 | 0.5 |
| hopper (broken joints) | 75% | 0.5 | 0.1 | 2.0 |
| hopper (big head) | 10% | 0.5 | 0.1 | 1.0 |
| walker (broken right foot) | 75% | 2.0 | 0.1 | 1.0 |
| walker (no right thigh) | 75% | 5.0 | 0.1 | 1.0 |

Table 3: Hyperparameter settings.

| Hyperparameter | Value |
|---|---|
| **Shared** | |
| Actor network | (256, 256) |
| Critic network | (256, 256) |
| Batch size | 256 |
| Learning rate | $3 \times 10^{-4}$ |
| Optimizer | Adam (Kingma, 2014) |
| Discount factor | 0.99 |
| Replay buffer size | $10^6$ |
| Warmup steps | 0 for DADiff and PAR, $10^5$ for others |
| Activation function | ReLU (Nair & Hinton, 2010) |
| Target update rate | $5 \times 10^{-3}$ |
| Temperature coefficient | 0.2 |
| Target domain interaction frequency | 10 |
| **DARC, SAC-IW** | |
| Classifier network | (256, 256) |
| **PAR** | |
| Encoder network | (256, 256) |
| Latent dimension | 256 |
| **VGDF** | |
| Dynamics model | (256, 256) |
| Ensemble size | 7 |
| Data selection ratio | 25% |
| **DADiff** | |
| Noise model | (256, 256) |
| Diffusion timesteps | 100 |
| Beta scheduler | Cosine scheduler (Nichol & Dhariwal, 2021) |

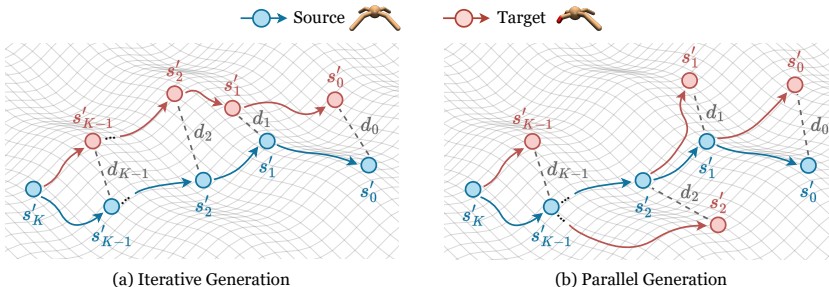

(a) Iterative Generation  (b) Parallel Generation

Figure 7: Generation forms of diffusion models to estimate the dynamics discrepancy. (a) Iterative generation form. The target-domain latent states are generated iteratively from $s'_K$ to $s'_0$, leading to more computational cost. (b) Parallel generation form. The target-domain latent states are generated in parallel based on the previous source-domain latent states, which is more efficient.

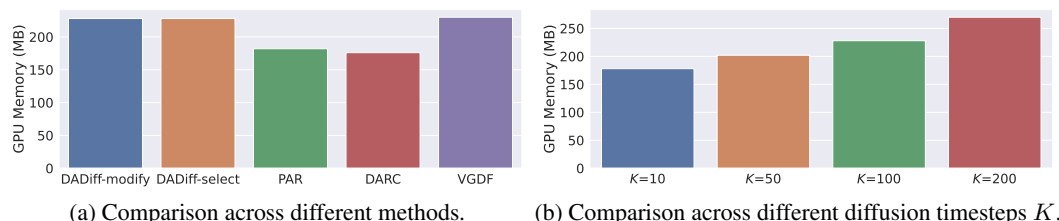

(a) Comparison across different methods.  (b) Comparison across different diffusion timesteps $K$.

Figure 8: GPU memory cost comparison on *halfcheetah (broken back thigh)* task. (a) The GPU memory cost of our method and VGDF is slightly higher than other baselines. (b) With the increase of diffusion timesteps $K$, the GPU memory cost increases slightly.

## F  EXTENDED EXPERIMENTAL RESULTS

In this section, we provide more experimental results, including extended computational cost analysis, extended results on stochastic environments, extended parameter studies, and extended reward distribution.

### F.1  EXTENDED COMPUTATIONAL COST ANALYSIS

In this section, we further analyze the computational cost of our proposed method. We conduct experiments on *halfcheetah (broken back thigh)* task and report the GPU memory cost of our method and baselines in Figure 8a. The results demonstrate that our method and VGDF incur slightly higher GPU memory consumption than other baselines. Compared to PAR and DARC, the additional GPU memory cost of our method mainly comes from the process of generating latent states. Since, unlike a full reverse diffusion process that sequentially generates target-domain next states, our method measures the discrepancy between source and target domains by evaluating multiple latent states in parallel, which leads to a slight increase in GPU memory cost. Meanwhile, the overall training time remains comparable to baseline methods, as shown in Figure 3, Section 5.2.

In addition, as the latent states in the generative trajectory are estimated in parallel, additional GPU memory cost would be related to the diffusion timesteps $K$. We conduct experiments on *halfcheetah (broken back thigh)* task with different diffusion timesteps $K$ and report the GPU memory cost in Figure 8b. We find that the GPU memory cost increases with $K$.

### F.2  EXTENDED PARAMETER STUDIES

We provide additional results on the parameter studies of penalty coefficient $\lambda$ and data selection ratio $\xi$ in Figure 9 and Figure 10, respectively. We raise the same conclusions as in Section 5.3, *i.e.*, the optimal value of penalty coefficient $\lambda$ and data selection ratio $\xi$ is task-dependent, and proper choices of these two hyperparameters can lead to better adaptation performance. In addition, we

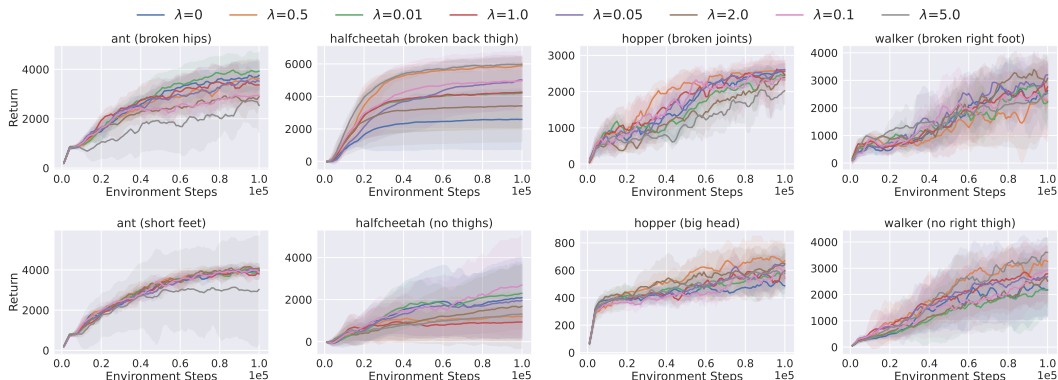

Figure 9: Extended parameter study of DADiff-modify on penalty coefficient $\lambda$. The solid curves and the shaded regions denote the mean and standard deviation over five random seeds, respectively.

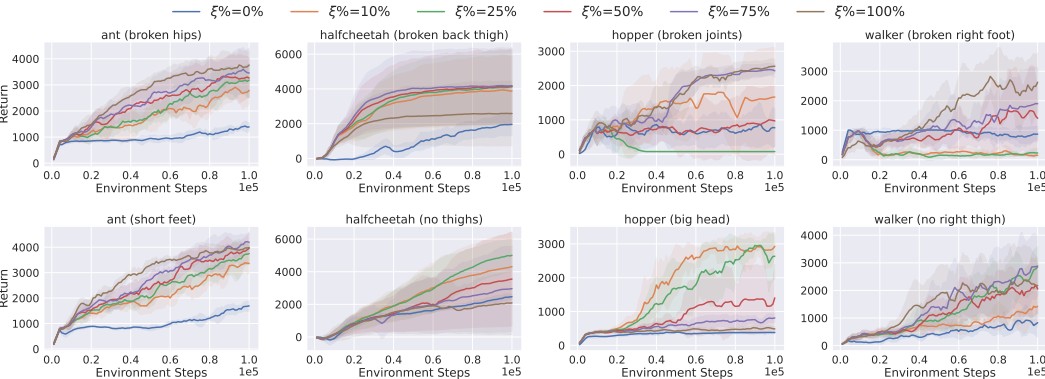

Figure 10: Extended parameter study of DADiff-select on penalty coefficient $\xi\%$. The solid curves and the shaded regions denote the mean and standard deviation over five random seeds, respectively.

must acknowledge that it shows a comparable performance in some tasks when reward modification or data selection is not applied, but it does not undermine the effectiveness of our method, as the performance can be further improved with appropriate hyperparameter settings.

We also provide additional results of DADiff-select on the parameter studies of target domain interaction frequency $F$ and diffusion timesteps $K$ in Figure 11a and Figure 11b, respectively. We find a different conclusion from DADiff-modify, *i.e.*, the adaptation performance of DADiff-select does not reach a plateau with the increase of target domain interaction frequency $F$ or diffusion timesteps $K$ in some tasks. But for the uniformity of our method, we set $F = 10$ and $K = 100$ in all tasks.

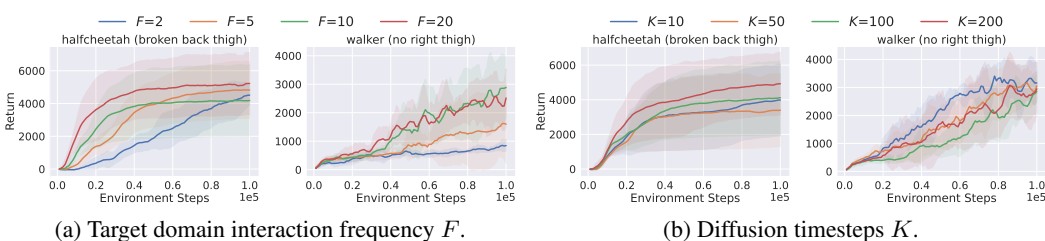

(a) Target domain interaction frequency $F$.    (b) Diffusion timesteps $K$.

Figure 11: Extended parameter study of DADiff-select on target domain interaction frequency $F$ and diffusion timesteps $K$. The solid curves and the shaded regions denote the mean and standard deviation over five random seeds, respectively.

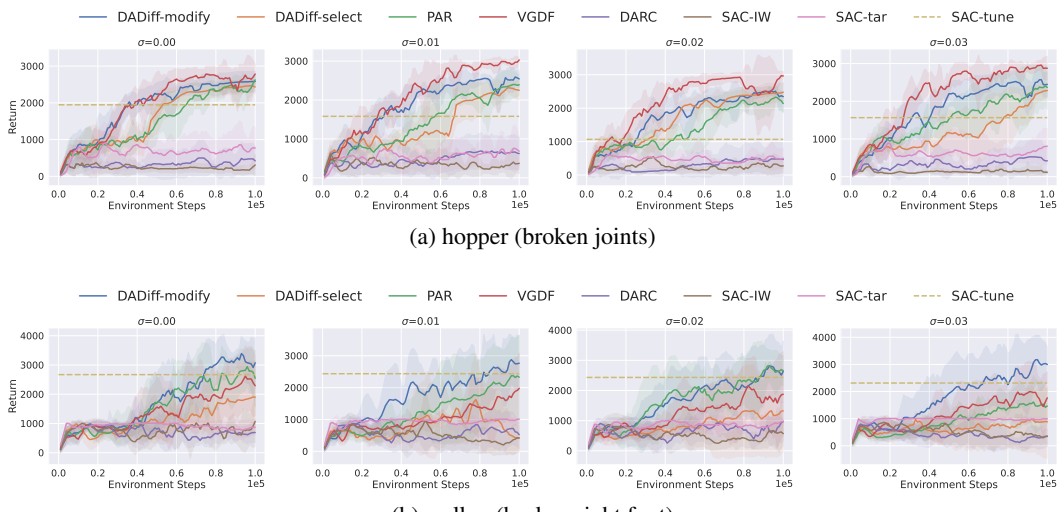

(a) hopper (broken joints)

(b) walker (broken right foot)

Figure 12: Adaptation performance with stochastic dynamics. The solid curves and the shaded regions denote the mean and standard deviation over five random seeds, respectively.

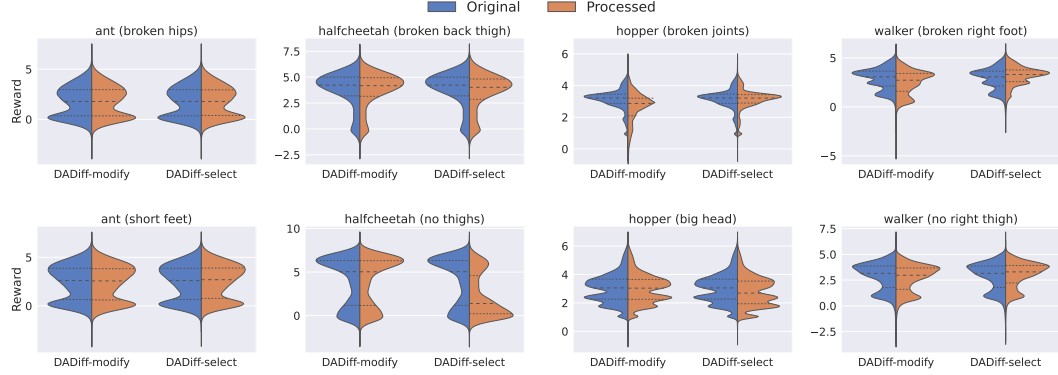

Figure 14: Extended reward distribution of DADiff-modify and DADiff-select. "Original" denotes the source-domain reward distribution prior to processing, whereas "Processed" denotes it after modification or selection.

### F.3 EXTENDED RESULTS ON STOCHASTIC ENVIRONMENTS

We first provide the details of the stochastic environments used in our experiments. Specifically, we define a Gaussian mixture model, which consists of two Gaussian components, to introduce stochasticity into the environment dynamics. The two components are $\mathcal{N} \sim (-0.1, \varsigma^2)$ with weight 0.7, and $\mathcal{N} \sim (0.1, \varsigma^2)$ with weight 0.3. An example with $\varsigma = 0.01$ is illustrated in Figure 13. Based on this Gaussian mixture model, we add the sampled noise to the action $a$ at each timestep during the interaction with the target environment. We provide more experimental results on *hopper (broken joints)* and *walker (broken right foot)* tasks in Figure 12. The results demonstrate that DADiff-modify performs the best among all methods on *walker (broken right foot)* task and the second best on *hopper (broken joints)* task.

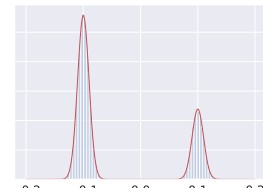

Figure 13: An example of the Gaussian mixture model with $\varsigma = 0.01$.

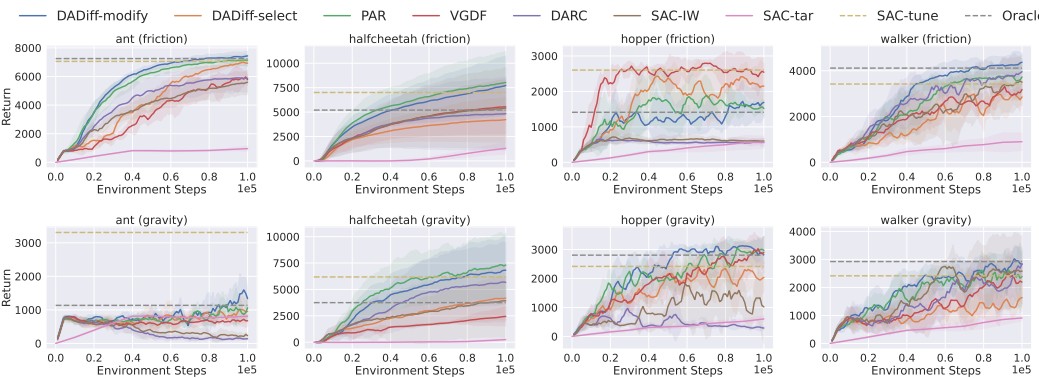

Figure 15: Adaptation performance on friction (top) and gravity (bottom) shifts. The solid curves and the shaded regions denote the mean and standard deviation over five random seeds, respectively. DADiff demonstrates superior or highly competitive performance against all baselines in the majority of tasks.

### F.4 Extended Reward Distribution

We provide additional results on the reward distribution of DADiff-modify and DADiff-select in Figure 14. We find that DADiff-modify only slightly modifies the source domain reward distribution in most tasks, as the reward penalty is small in most cases. On the contrary, DADiff-select tends to change the source domain reward distribution more significantly. In most tasks, more low-reward data helps polices to avoid learning from harmful transitions in the source domain and thus improves the adaptation performance.

### F.5 Extended Adaptation Performance Evaluation

**Friction and gravity shifts.** We follow the experimental settings of ODRL Lyu et al. (2024c) and provide additional experimental results on another eight tasks with friction and gravity shifts (shift level 0.5) in Figure 15. Our method achieves better performance in 5 out of 8 tasks and comparable performance in the remaining tasks. Our method surpasses Oracle in all tasks and achieves comparable performance to SAC-tune in most tasks. We also find that reward modification methods, *e.g.*, PAR and DADiff-modify, consistently perform better in most tasks with friction and gravity shifts.

**Action mask.** To investigate the potential limitations of reward modification methods, we follow the experimental setups in DARAIL (Guo et al., 2024) and set the value of 0-index in the action of the source domain frozen to 0. We provide the results of existing reward modification methods and Oracle, which is the SAC algorithm trained in the target domain for 1M environmental steps, in Figure 16. We find that DARC fails in such tasks, while PAR and our method perform well and can achieve the optimal return near the performance of Oracle in the target domain. It indicates that improving the theoretical analysis of reward modification methods can achieve the optimal performance in the target domain as well.

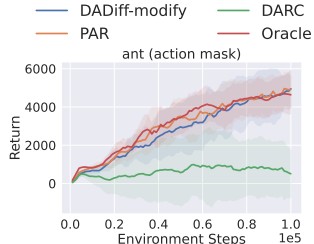

Figure 16: Adaptation performance under a broken source environment setting. Both DADiff-modify and PAR achieve similar performance to Oracle, while DARC fails.

### F.6 Extended Reward Penalty Analysis

We further quantify the reward penalty measured by PAR and DADiff-modify in the *halfcheetah (no thighs)* and *hopper (big head)* tasks to have a better understanding of our method in Figure 17. We find that our method always provides a lower reward penalty than PAR and corrects the reward with less effect. It can make our method benefit from such low penalties and use the small but

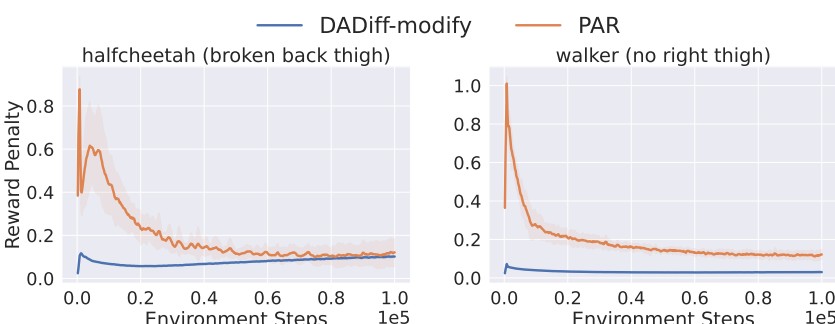

Figure 17: Reward penalty comparison between PAR and DADiff-modify. The solid curves and the shaded regions denote the mean and standard deviation over five random seeds, respectively. Our method always provides a lower reward penalty compared to PAR.

critical penalties to achieve similar or even better performance compared to PAR, demonstrating the advantage of a more fine-grained manner.

### F.7 EXTENDED VARIANTS ABLATION STUDY

Table 4: Variants ablation study. The mean and standard deviation are reported over five random seeds. The results demonstrate that each mechanism contributes differently to different dynamics adaptation tasks.

| Method | hopper (big head) | walker (broken right foot) |
|---|---|---|
| DADiff-modify | $701.09 \pm 133.52$ | $3390.44 \pm 959.41$ |
| DADiff-select | $2935.83 \pm 1033.59$ | $1905.88 \pm 436.35$ |
| DADiff-modify & select | $2625.00 \pm 786.94$ | $1977.73 \pm 414.43$ |

We conduct an extended ablation study to further examine how the reward modification and data selection mechanisms affect performance in the *halfcheetah (no thighs)* and *hopper (big head)* tasks in Table 4. In the *halfcheetah (no thighs)* task, DADiff-select outperforms DADiff-modify, whereas the opposite holds in the *hopper (big head)* task. We combine both mechanisms and denote this variant as DADiff-modify&select. The results demonstrate that DADiff-modify&select will lead to intermediate performance, lying between the two variants, which is consistent with our previous finding that each mechanism contributes differently to different dynamics adaptation tasks.