# OpenReview forum: "Leveraging Generative Trajectory Mismatch for Cross-Domain Policy Adaptation"
_ICLR.cc/2026/Conference — Submitted to ICLR 2026_

### Official Review · Reviewer_azow · 2025-10-15

**Soundness:** 3
**Presentation:** 3
**Contribution:** 3
**Rating:** 6
**Confidence:** 4

**Summary:**

This paper introduces DADiff, a diffusion-based framework that addresses the challenge of transferring reinforcement learning policies across domains with different dynamics. By leveraging the generative trajectory discrepancy between source and target domains, DADiff estimates dynamics mismatch and adapts policies through either reward modification or data selection strategies. Supported by theoretical analysis showing the performance difference is bounded by generative deviation, the method demonstrates superior effectiveness in experiments with kinematic and morphology shifts compared to existing approaches.

**Strengths:**

# Strengths

- This paper is well-motivated and mostly well-written
- This paper is easy to follow, and the studied topic is of importance in the context of the reinforcement learning community. It is always important to develop more general and stronger transfer algorithms in RL, especially considering the fact that online off-dynamics RL papers have rarely appeared in recent years
- The authors include theoretical analysis to provide better guarantees for the proposed method (despite the fact that some of the theoretical results resemble those in prior works, they are still interesting and bring some insights into the cross-domain reinforcement learning). I appreciate that the authors include a detailed discussion about the connections between the theoretical bounds of their method and those of PAR
- The presentation is good, and I like the way the authors tell the whole story
- The parameter study is extensive, covering numerous tasks in the main text and the appendix.

**Weaknesses:**

# Weaknesses

- The authors propose to address the online policy adaptation problem from the perspective of generative modeling; however, the downstream methods still rely on reward modification or data filtering, which resembles DARC, PAR, and VGDF
- The evaluations are limited to kinematic shift and morphology shift. As far as the reviewer can tell, ODRL provides other dynamics shifts like gravity shift, friction shift, etc. This paper can benefit greatly from extending its experimental scope
- The authors mention flow matching in the main text. This raises questions that there are numerous generative modeling methods other than diffusion models. This paper lacks a comparison between different generative modeling methods.

Overall, I would recommend a "weak accept" of this paper.

**Questions:**

# Questions

1. As a generative modeling method, diffusion can also be used for data augmentation, e.g., generating samples that lie in the scope of the target domain. What is the insight in using diffusion model for *Generative Trajectory Mismatch* rather than target domain data augmentation?
2. How diffusion models compare against other generative modeling methods like flow matching, VAE?
3. The diffusion steps seem to have a significant impact on DADiff. Can authors provide more insights on how to select this parameter and why different diffusion steps can have such significant impacts on DADiff?

---

> ### Author Response · Authors · 2025-11-22
> **Author's Response (1/2)**
>
> We are thankful for your time and thoughtful feedback, especially related to the discussions on the generative modeling perspective and benchmark coverage. We are glad that you found our paper well-motivated, clearly written, and appreciated its theoretical analysis, presentation, and thorough parameter study. To facilitate the review process, we summarize our main clarifications and updates as follows:
> - We clarify that our work focuses on fine-grained, distributional measurement of generative trajectory discrepancy within the reward-modification and data-selection framework, and we will explore broader adaptation approaches in future work.
> - We add extended experiments on gravity and friction shifts in Appendix F.5, where our method achieves better or comparable performance in 5 out of 8 tasks, further validating its effectiveness.
> - We provide a CVAE-based implementation in Appendix C and show that DADiff consistently outperforms other generative variants, indicating its stronger capacity to model generative trajectory deviation.
> - We clarify that our method focuses on quantifying source–target dynamics discrepancy rather than performing data augmentation, effectively leveraging abundant source-domain data for fine-grained trajectory-level adaptation while remaining efficient through parallel generation.
> - We fix $K=100$ in our experiments and find performance depends on various factors; we plan to explore adaptive or variants with fewer hyperparameters in future work.
>
> Please find the point-by-point responses to the reviewer's comments below.
>
> **[W1] Practical algorithm implementation**
> > The authors propose to address the online policy adaptation problem from the perspective of generative modeling; however, the downstream methods still rely on reward modification or data filtering, which resembles DARC, PAR, and VGDF.
>
> Thanks for your comments. We would like to mention that there are various approaches to address the policy adaptation problem. Reward modification and data filtering are two of them, which are still widely used in multiple works [1, 2]. **Our work focuses on achieving better performance via measuring generative trajectory discrepancy in a more fine-grained and distributional manner**, and further investigates the connection between PAR and our method. We will explore more different approaches to address the policy adaptation problem in the future. Thanks again for your suggestions.
>
> **[W2] Extended adaptation experiments**
> > The evaluations are limited to kinematic shift and morphology shift. As far as the reviewer can tell, ODRL provides other dynamics shifts like gravity shift, friction shift, etc. This paper can benefit greatly from extending its experimental scope
>
> Thanks for your suggestions on extended experiments. To better demonstrate the effectiveness of our method, **we introduce extended adaptation experiments on gravity shift and friction shift (shift level 0.5) provided by ODRL [3] in Appendix F.5**, which are included in the latest version of paper. We find that our method achieves better performance in 5 out of 8 tasks and comparable performance in the remaining tasks, which is consistent with our previous experimental results.
>
> **[W3 & Q2] Comparison among various generative models**
> > The authors mention flow matching in the main text. This raises questions that there are numerous generative modeling methods other than diffusion models. This paper lacks a comparison between different generative modeling methods. How diffusion models compare against other generative modeling methods like flow matching, VAE?
>
> Thanks for your valuable suggestions. **We further provide another practical implementation of our method based on CVAE [4] in Appendix C.** The results show that diffusion-based implementation (DADiff-modify) achieves the best performance across different implementations, which indicates that diffusion models are the better choice to model the generative trajectory deviation. The failure of CVAE-based method might be caused by the coarse generation ability.

---

> ### Author Response · Authors · 2025-11-22
> **Author's Response (2/2)**
>
> **[Q1] Discussion on generative trajectory modeling**
> > As a generative modeling method, diffusion can also be used for data augmentation, e.g., generating samples that lie in the scope of the target domain. What is the insight in using diffusion model for Generative Trajectory Mismatch rather than target domain data augmentation?
>
> Thanks for your insightful thoughts on the policy adaptation problem. Indeed, diffusion models can be used both for generating trajectories to measure mismatch and for augmenting target domain data. The key insight of using generative trajectory mismatch lies in its direct focus on quantifying the discrepancy between domains, which cannot be directly obtained through standard data augmentation. While data augmentation generates additional target-domain samples, it does not provide a measure of source-target dynamics mismatch, making it cannot use plenty of source-domain data. In contrast, generative trajectory modeling leverages source-domain data to provide a direct, trajectory-level signal of dynamics discrepancy, enabling more effective and targeted adaptation.
>
> In addition, generative trajectory modeling in our parallel generation form is also more efficient than diffusion-based data augmentation. Since sequentially generating target-domain samples requires iterative sampling across all steps, incurring $K$-fold time complexity, which is particularly costly in online interactions. Our parallel approach mitigates this issue, making it suitable for online policy adaptation tasks.
>
> **[Q3] Discussion on diffusion steps**
> > The diffusion steps seem to have a significant impact on DADiff. Can authors provide more insights on how to select this parameter and why different diffusion steps can have such significant impacts on DADiff?
>
> We appreciate your thoughtful comments. In our experiments, we fix diffusion timesteps to 100 ($K=100$). According to the parameter study of diffusion timesteps, we also observed that the performance of our method is affected by diffusion timesteps. However, the optimal diffusion timesteps depend on the difficulty of tasks and many other factors. We were working to find out the potential regularity of the optimal diffusion timesteps. But there are too many factors that could have an influence on the performance. Therefore, we suggest directly adopting diffusion timesteps $K$ of 100. We will further explore the potential regularity and develop methods with fewer hyperparameters.
>
>
>
> **References**
>
> [1] Lyu, Jiafei, et al. "Cross-Domain Offline Policy Adaptation with Optimal Transport and Dataset Constraint." *The Thirteenth International Conference on Learning Representations*. 2025.
>
> [2] Kong, Lingkai, et al. "Composite Flow Matching for Reinforcement Learning with Shifted-Dynamics Data." *Advances in Neural Information Processing Systems* 38 (2025).
>
> [3] Lyu, Jiafei, et al. "ODRL: A Benchmark for Off-Dynamics Reinforcement Learning." *Advances in Neural Information Processing Systems* 37 (2024): 59859-59911.
>
> [4] Kingma, Diederik P., and Max Welling. "Auto-encoding variational bayes." arXiv preprint arXiv:1312.6114 (2013).

---

> > ### Comment · Reviewer_azow · 2025-11-25
> >
> > Thank you for the rebuttal. I keep my rating and recommend a weak accept of this paper.
> >
> > A minor point, there are always too many curves in one figure (e.g., Figure 6), it can be a bit hard to interpret the figure. The authors are encouraged to find better ways of showing the comparison results, maybe in the final version.

---

> > > ### Author Response · Authors · 2025-11-27
> > > **Author's Response**
> > >
> > > We sincerely thank the reviewer for maintaining a positive assessment and recommending a weak accept of our paper. We truly appreciate your thoughtful comments and constructive suggestions, which have greatly helped us further refine the paper.
> > >
> > > We agree that clearer results organization could enhance readability, and the quantitative results of Figure 6 are further provided in the table below, which indicates that diffusion models can achieve near-Oracle performance. We will take the suggestion into account in the next version to ensure smoother comparison and overall presentation quality.
> > >
> > > Quantitative results of Figure 6:
> > > | Method | halfcheetah (broken back thigh) | walker (no right thigh) |
> > > |--|--|--|
> > > | SAC-tune | 4018.14 $\pm$ 3017.84 | 2589.63 $\pm$ 617.44 |
> > > | VGDF | 4163.27 $\pm$ 1159.69 | 3271.66 $\pm$ 1055.00 |
> > > | PAR | 4136.08 $\pm$ 1308.62 | 2766.03 $\pm$ 763.77 |
> > > | DAVAE-modify | 4189.52 $\pm$ 1259.52 | 3502.15 $\pm$ 1129.63 |
> > > | DAFlow-modify | 5043.95 $\pm$ 1466.50 | 3208.75 $\pm$ 1032.73 |
> > > | DADiff-select | 4189.62 $\pm$ 1253.17 | 2885.38 $\pm$ 862.12 |
> > > | DADiff-modify | **5882.54** $\pm$ 1753.20 | **3614.90** $\pm$ 1103.69 |
> > > |--|--|--|
> > > | Oracle | 5849.29 $\pm$ 103.28 | 4098.48 $\pm$ 199.24 |
> > >
> > >
> > > We are also grateful that the reviewer recognizes the motivation, theoretical soundness, and empirical contributions of our work. We have carefully addressed all raised concerns and will further refine the clarity and completeness of the paper.
> > >
> > > We sincerely hope that these improvements can convey the full value and potential impact of our approach in the final evaluation. Thank you again for your thoughtful feedback and encouraging assessment.

---

### Official Review · Reviewer_w6iz · 2025-10-30

**Soundness:** 3
**Presentation:** 1
**Contribution:** 2
**Rating:** 4
**Confidence:** 3

**Summary:**

This paper proposes DADiff, an online dynamics adaptation method for RL that measures source–target dynamics shift via generative trajectory deviation from diffusion models. It developed two variants: reward modification and data selection. A performance bound links return gap to KL terms along a shared latent trajectory. Experiments on 4 MuJoCo envs report competitive or superior performance to DARC, VGDF, PAR, SAC‑tune, and SAC‑tar.

**Strengths:**

1. Clear theoretical link from generative trajectory discrepancy to performance, with clean proof and recovery of PAR as a special case.
2. Consistent improved empirical performance on many tasks. DADiff‑modify often leads; DADiff‑select is strong when penalties mis‑shape rewards.
3. Parallel latent sampling avoids reverse‑chain cost; runtime comparable to model‑free baselines and far below VGDF.

**Weaknesses:**

1. Baseline fairness. SAC‑tar is trained for 10^5 target steps, while DADiff and SAC‑tune use 1M source steps + 10^5 target steps. This probes a target‑only‑from‑scratch regime but does not compute‑match total experience. Please add a compute‑matched target‑only control with comparable total environment interactions and gradient updates

2. Insufficient analysis: The text narrates Fig. 2 but provides little analysis in Sec 5.2. Please also quantify the deviation differences between the two generative trajectories, since the paper only covers the computational effeiciency. There should exist a deviation difference between these two trajectories.

3. Writing quality: Multiple typos, symbol switches, and undefined or late‑defined notation reduce clarity. For examples: Fig 4(a) using $\gamma$ while Eqn 11 using $\lambda$. $\phi_i$ is undefined in Eqn 14 until I found out the algorithm is based on SAC. Sec 5.3 states optimal $\lambda$ is task-dependent while Sec E.2 (line 1019) says $\lambda$ is task-independent.

**Questions:**

Same as weakness

Additional questions:
1. Is there a missing square in the Eqn 12 and 13? If not, justify using $E[Q−TQ]$ rather than MSE. If yes, re‑run results with corrected losses and report any deltas.
2. Can you extend the analysis of why "directly filtering for transitions with low dynamics mismatch is a more effective strategy than modifying rewards." in your Sec 5.2 (line 352). Provide mechanism‑level reasoning and ablations that include filtering only vs reward‑shaping only vs both. Maybe analysis from the perspective of probabilistic trajectory in diffusion model could explain why filtering is better.

---

> ### Author Response · Authors · 2025-11-22
> **Author's Response (1/2)**
>
> We are thankful for your time and constructive feedback, especially related to the discussions on baseline fairness and trajectory-level analysis. We were glad to hear that you found our paper provides a clear theoretical link between generative trajectory discrepancy and policy performance, with consistent empirical improvements across tasks and strong computational efficiency. To facilitate the review process, we summarize our main clarifications and updates as follows:
> - We provide a compute-matched SAC trained in the target domain for 1M steps (Oracle) in Figure 2, showing that our method achieves comparable performance to Oracle in certain tasks.
> - We measure reward penalties (deviation) measured by PAR and DADiff-modify in Appendix F.6, demonstrating that our method achieves lower penalties and better performance in a fine-grained manner.
> - All typos, missing symbols, and notation inconsistencies have been corrected, and the results remain valid with MSE loss as implemented.
> - We provide mechanism-level analysis and ablation experiments in Appendix F.7, showing that data selection or reward modification performs better individually, while combining both yields intermediate performance.
>
> Our detailed point-by-point responses to the reviewer’s comments are provided below.
>
> **[W1] Extended baselines**
> > SAC‑tar is trained for 10^5 target steps, while DADiff and SAC‑tune use 1M source steps + 10^5 target steps. This probes a target‑only‑from‑scratch regime but does not compute‑match total experience. Please add a compute‑matched target‑only control with comparable total environment interactions and gradient updates
>
> Thanks for your suggestion. In our paper, we follow the baseline settings in PAR. To better compare the results of different methods and the optimal results achieved in the target domain, we further provide the results of SAC algorithm trained in the target domain for 1M environmental steps. **We add these results in Figure 2, denoted as Oracle.** Our method achieves comparable performance to Oracle in certain tasks, which demonstrates the effectiveness of our method.
>
> **[W2] Deviation quantification analysis**
> > The text narrates Fig. 2 but provides little analysis in Sec 5.2. Please also quantify the deviation differences between the two generative trajectories, since the paper only covers the computational effeiciency. There should exist a deviation difference between these two trajectories.
>
> Thanks for highlighting the importance of deviation quantification analysis. We highly agree that deviation quantification can lead to a better understanding for our method. **We provide the reward penalty (deviation) measured by PAR and DADiff-modify in Appendix F.6.** We find that our method often has a lower penalty than PAR and can correct the reward with less effect. Therefore, our method can benefit from such low penalties and use the small but critical penalties to achieve similar or even better performance compared to PAR, demonstrating the advantage of a more fine-grained manner.
>
>
> **[W3 & Q1] Writing quality improvement**
> > Multiple typos, symbol switches, and undefined or late‑defined notation reduce clarity.
>
> Thanks for your comments. We have corrected the writing errors mentioned by reviewers. According to the results provided in Section 5.3, we reclarify that the optimal $\lambda$ is task-dependent to avoid misunderstanding.
>
> Besides, **the missing square in Equation 12 and 13 is actually a writing error.** We adopted MSE loss in our implementation and all of the results based on this loss are correctly provided in the paper. Therefore, there is no additional result that needs to be provided.
>
> We sincerely thank the reviewer for pointing out these typos, and we have corrected these errors in the latest version of our paper. We will check for other possible errors.

---

> ### Author Response · Authors · 2025-11-22
> **Author's Response (2/2)**
>
> **[Q2] Extended analysis for additional variant**
> > an you extend the analysis of why "directly filtering for transitions with low dynamics mismatch is a more effective strategy than modifying rewards." in your Sec 5.2 (line 352). Provide mechanism‑level reasoning and ablations that include filtering only vs reward‑shaping only vs both. Maybe analysis from the perspective of probabilistic trajectory in diffusion model could explain why filtering is better.
>
> Thanks for your insightful suggestions. First, we reclarify the conclusion - "in certain tasks, directly filtering for transitions with low dynamics mismatch is a more effective strategy than modifying rewards" - to avoid misunderstanding. **We get this conclusion just based on the results of `halfcheetah (no thighs)` and `hopper (big head)` provided in Figure 2.** In both tasks, VGDF and DADiff-select achieve significantly superior performance compared to other methods, demonstrating the effectiveness of data selection methods in certain tasks. We provided an analysis from the reward distribution perspective, which shows that DADiff-select generates a higher distribution in the low-reward region than DADiff-modify in both tasks. It suggests that low-reward data may play an important role in these tasks, guiding the policy to avoid undesirable states and actions.
>
> In addition, we further introduce an additional variant with both mechanisms in order to figure out the potential reason for this phenomenon at the mechanism level. **We conduct experiments on `hopper (big head)` and `walker (broken right foot)` tasks, where the data selection variant and reward modification variant achieve better performance, respectively, as shown below and in Appendix F.7.** The results demonstrate that combining both mechanisms will only lead to intermediate performance, lying between the two variants, which is consistent with our previous finding that each mechanism contributes differently to different dynamics adaptation tasks.
>
> Results on mechanism-level ablations:
> | Method | hopper (big head) | walker (broken right foot) |
> | --- | --- | --- |
> | DADiff-modify | 701.09 $\pm$ 133.52 | 3390.44 $\pm$ 959.41 |
> | DADiff-select | 2935.83 $\pm$ 1033.59 | 1905.88 $\pm$ 436.35 |
> | DADiff-modify & select | 2625.00 $\pm$ 786.94 | 1977.73 $\pm$ 414.43 |
>
>
> To figure out the effectiveness of our method from the generative trajectory perspective, we provide the deviation quantification analysis in [W2], which indicates that our method can use the small but critical penalties to achieve similar or even better performance compared to PAR.
>
> We are delighted to have further discussions if more clarifications are needed.

---

### Official Review · Reviewer_5rqK · 2025-10-31

**Soundness:** 2
**Presentation:** 3
**Contribution:** 2
**Rating:** 2
**Confidence:** 4

**Summary:**

This paper provides a diffusion-based method to obtain the domain gap and provides a reward modification and data filtering method for policy learning. They provide a theoretical guarantee of the policy $\pi$'s performance on the two domain.  Theoretical results and empirical results shows performance improvement of the method.

**Strengths:**

The paper is well written and easy to follow. They propose a new diffusion-based domain gap measure method similar to DARC and PAR. Similar to DARC and PAR, they identify a performance gap in policy $\pi$ between the source and target domains， defined by the KL divergence of the latent representation.

**Weaknesses:**

1, the odrl benchmark has both gravity shift and friction shift, which are not included in the experiments. Also, what is the shift level of the experiment? Is it easy, medium or hard?

2, the novelty of the paper seems not significant to me. The high-level idea of it is to obtain a more fine-grained shift measurement compared to DARC and PAR, and the theoretical analysis is actully similar to those paper, except with sligtly different notation of the shfit measurement. Also, the performance doesn't have a significant improvement compared to them.

3, DARC and PAR have been shown to be ineffective when the shift is large. What is your performance on a large shift case?

4, The performance of your method seems to rely on an assumption that the domain shift is not that large. If the shift is too large, the KL in Eq. 5 will grow extremely large, or even infinity. The performance is bounded only when the KL of the source and target is bounded. Also, as stated in [1], the KL can be ill defined when the shift is large because there is no support of some target transition in source domain.

In summary, I am questioning whether the reward modification method is still a valid method to solve the off-dynamcis RL problem as many previous work [1,2] has shown the limitation of it and when the shift is large (the joint distribution is small), the reward modicication method always fails, performing good in the source but poorly in the target.

[1] Composite Flow Matching for Reinforcement Learning with Shifted-Dynamics Data
[2] Off-Dynamics Reinforcement Learning via Domain Adaptation and Reward Augmented Imitation

**Questions:**

See weakness.

---

> ### Author Response · Authors · 2025-11-22
> **Author's Response (1/4)**
>
> We sincerely appreciate the reviewer’s valuable and thoughtful insights, especially related to the discussions on the benchmark coverage and theoretical validity under large domain shifts. We are also grateful for your positive remarks on the clarity and readability of our paper. But it seems that some aspects of existing policy adaptation methods might be misunderstood, and we would like to provide clarifications below in a point-by-point manner. To facilitate the review process, we summarize our main clarifications and updates as follows:
> - We clarify that all experiments follow the PAR setting and add extended adaptation results on gravity and friction shifts in Appendix F.5, where our method outperforms others in 5 out of 8 tasks.
> - We emphasize that our method goes beyond a notational modification by introducing a fine-grained, distributional formulation of dynamics discrepancy that relaxes the restrictive assumptions in DARC and PAR, achieving consistent improvements in 6 of 8 tasks and superior robustness in stochastic environments.
> - Following DARAIL’s large-shift setting, we show that our method reaches near-Oracle performance, while DARC fails, demonstrating strong robustness under large domain shifts.
> - We clarify that our formulation relies on latent state transitions rather than physical ones, effectively avoiding divergence issues when source-domain probabilities vanish, and we support this claim with large-shift experiments in Appendix F.5.
>
> **[W1] Experimental setups and extended adaptation experiments**
>
> > the odrl benchmark has both gravity shift and friction shift, which are not included in the experiments. Also, what is the shift level of the experiment? Is it easy, medium or hard?
>
>
>
> Thanks for your suggestions on clearer experimental setups and extended experiments. We followed the experimental settings of PAR, which adopts the standard domain-shift configurations without assigning explicit labels such as “easy”, “medium”, or “hard” for each task.
>
> As suggested, we implemented all algorithms using the official ODRL codebase [1] and conducted additional adaptation experiments under the ODRL gravity-shift and friction-shift settings (shift level 0.5). The extended results are presented in Appendix F.5 of the updated version. We observe that our method achieves the best performance on 5 out of 8 tasks and ranks among the top two methods on the remaining tasks, which is consistent with our findings in the main experiments.
>
> We would be happy to provide further experiments or analyses if additional clarification is needed.

---

> ### Author Response · Authors · 2025-11-22
> **Author's Response (2/4)**
>
> **[W2] Novelty and performance concerns**
> > the novelty of the paper seems not significant to me. The high-level idea of it is to obtain a more fine-grained shift measurement compared to DARC and PAR, and the theoretical analysis is actully similar to those paper, except with sligtly different notation of the shfit measurement. Also, the performance doesn't have a significant improvement compared to them.
>
> Thanks for your comments. Our method is actually inspired by DARC and PAR, which derive the performance bound from domain divergence perspective, but it goes beyond a mere notational modification. While DARC and PAR mainly measure coarse distinctions between domains, our method can capture the dynamics discrepancy in a more fine-grained and distributional manner. We provide a detailed discussion to point out the practicality and novelty of our method:
>
> - **DARC vs. DADiff.** The theoretical analysis of DARC is built upon an assumption that *every transition with non-zero probability in the target domain has non-zero probability in the source domain*. The practical algorithm of DARC utilizes two classifiers to predict whether a transition came from the source or target domain and measure $\log \frac{p_\mathrm{tar}(s^\prime|s,a)}{p_\mathrm{src}(s^\prime|s,a)}$, which could lead to failure when $p_\mathrm{src}(s'|s,a)$ is approaching to 0. However, our method is based on a theoretical analysis concerning latent state transitions $P_M(s'_{k-1}|s'_k,s,a)$ instead of physical transitions $P_M(s'|s,a)$. **This kind of theory could avoid direct utilization of physical transitions, which could lead to infinity when calculating KL divergence in DARC.** It can produce more appropriate reward penalties as well, as the experiments in Appendix F.6 and the paper of PAR presented. Therefore, compared to DARC, our method can handle more general tasks, especially when the transition probability in the source domain is approaching 0. We follow the experimental setup of DARAIL [2] and provide additional experiments on broken environments in Appendix F.5, which freeze the 0-index value in the action to 0 in the source-domain action, leading to a large shift and less support. The details are provided in [W3], which shows that our method can achieve a near-optimal performance while DARC fails.
> - **PAR vs. DADiff.** As we presented in Section 6, the theoretical analysis of PAR is built upon a one-to-one representation mapping assumption, which may not hold in practice, especially in stochastic environments. Meanwhile, PAR only measures a coarse distinction between domains via $D_{\mathrm{KL}}\left( P(z \mid s^\prime_{\mathrm{src}}) \|\| P(z \mid s^\prime_{\mathrm{tar}}) \right)$. **In contrast, our method does not rely on such an assumption and measure the deivation in a more fine-grained manner via $\sum_{k=1}^{K} D_{\mathrm{KL}}(P_{\mathrm{src}}(s^\prime_{k-1} \mid s^\prime_k, s, a) \|\| P_{\mathrm{tar}}(s^\prime_{k-1} \mid s^\prime_k, s, a))$, enabling it to generalize to a wider range of tasks.** We provided the experiments in environments with stochastic dynamics in Table 1, which demonstrate that our method maintains more robust performance than PAR in stochastic settings. Meanwhile, we further provide the reward penalty measured by PAR and our method in Appendix F.6. We find that our method often has a lower penalty than PAR and can correct the reward with less effect, indicating that our method can use the small but critical penalties to achieve similar or even better performance compared to PAR.
>
> **Concerning performance improvement,** we would like to mention that **our method achieves the best performance on 6 out of 8 tasks in the main experiments of Section 5.2, 5 out of 8 tasks in the extended experiments of Appendix F.5, and ranks among the top two methods on the remaining tasks**. This consistent advantage across tasks indicates that our improvement is stable and practically meaningful.

---

> ### Author Response · Authors · 2025-11-22
> **Author's Response (3/4)**
>
> **[W3] Experiments on a large-shift case**
> > DARC and PAR have been shown to be ineffective when the shift is large. What is your performance on a large shift case?
>
> Thanks for your valuable insights. As suggested, **we conduct experiments in a large-shift domain and provide the results in Appendix F.5.** In this experiment, we follow the experimental setting of DARAIL [2], the 0-index value in the action is frozen in the source domain (action mask). We find that PAR and our method achieve the optimal performance that is similar to Oracle (SAC trained in the target domain for 1M environmental steps), while DARC fails. **It indicates that reward modification methods can achieve optimal performance in the large-shift environment.**
>
> Besides, we also follow the similar experimental setting in ODRL [1] and CompFlow [3], and provide additional experiments on gravity shift and friction shift to demonstrate the effectiveness of our method in the online setting. We find that our method achieves better performance in 5 out of 8 tasks and comparable performance in the remaining tasks, which proves the effectiveness of our method in various tasks with different shifts. The details are provided in [W1].
>
> We are delighted to provide more experiments in the additional large-shift environments provided by the reviewer.

---

> ### Author Response · Authors · 2025-11-22
> **Author's Response (4/4)**
>
> **[W4 & Summary] Discussion on KL-based theoretical analysis**
> > The performance of your method seems to rely on an assumption that the domain shift is not that large. If the shift is too large, the KL in Eq. 5 will grow extremely large, or even infinity. The performance is bounded only when the KL of the source and target is bounded. Also, as stated in [1], the KL can be ill defined when the shift is large because there is no support of some target transition in source domain.
> >
> > In summary, I am questioning whether the reward modification method is still a valid method to solve the off-dynamcis RL problem as many previous work [1,2] has shown the limitation of it and when the shift is large (the joint distribution is small), the reward modicication method always fails, performing good in the source but poorly in the target.
>
>
> Thanks for your insightful thought on the theoretical analysis. We agree that for DARC, there are several limitations of KL-based theoretical analysis, as mentioned in citations [2, 3]. As noted in [W2], the assumption used in DARC makes it vulnerable in large-shift settings, because directly relying on physical transitions $P_M(s'|s,a)$ becomes unreliable when the source-domain transition probability approaches zero.
>
> Both PAR and our method address this issue by avoiding the direct utilization of the physical transitions to **avoid direct dependence on physical transitions. PAR measures dynamics deviation using representation embeddings**, whereas our method uses **noisy latent states in the generative trajectory** to characterize the discrepancy. Although both fall under the broader category of KL-based approaches, neither method requires evaluating KL divergence on physical transition distributions, which prevents the failure modes highlighted in the comment. Our approach further measures the discrepancy at a more fine-grained, distributional level through the latent state transition formulation in $\sum_{k=1}^{K} D_{\mathrm{KL}}(P_{\mathrm{src}}(s^\prime_{k-1} \mid s^\prime_k, s, a) \|\| P_{\mathrm{tar}}(s^\prime_{k-1} \mid s^\prime_k, s, a))$, which offers improved stability in complex tasks.
>
> To support these points, as mentioned in [W3], we conducted additional experiments on large-shift domains. The results show that our method continues to perform effectively under substantial domain shifts. In addition, the extended experiments on gravity shift and friction shift (provided in [W1]) demonstrate that our method achieves the best performance on 5 out of 8 tasks and competitive performance on the remaining tasks, which further validates its robustness across different shift types.
>
> We also note that the limitations of KL-based reward modification have been discussed in recent work. DARAIL [2] analyzes the degradation of DARC under large shifts, which aligns with our observations. PAR and our method aim to improve performance through better theoretical grounding without relying on imitation learning, while approaches such as CompFlow [3] share the insight that dynamics discrepancy can be modeled through generative representations. These methods are not mutually exclusive and reflect the field’s broader trend toward more flexible representations of domain shift.
>
> In the next version of the paper, we will make these points more explicit and include a clearer discussion under large domain shifts. We will also provide a more detailed comparison of discrepancy modeling between PAR and our method.
>
>
> **References**
>
> [1] Lyu, Jiafei, et al. "ODRL: A Benchmark for Off-Dynamics Reinforcement Learning." *Advances in Neural Information Processing Systems* 37 (2024): 59859-59911.
>
> [2] Guo, Yihong, et al. "Off-dynamics reinforcement learning via domain adaptation and reward augmented imitation." Advances in Neural Information Processing Systems 37 (2024): 136326-136360.
>
> [3] Kong, Lingkai, et al. "Composite Flow Matching for Reinforcement Learning with Shifted-Dynamics Data." Advances in Neural Information Processing Systems 38 (2025).

---

### Official Review · Reviewer_G9sM · 2025-11-02

**Soundness:** 3
**Presentation:** 3
**Contribution:** 3
**Rating:** 6
**Confidence:** 4

**Summary:**

This paper addresses the problem of online dynamics adaptation in reinforcement learning, where a policy is pre-trained in a source domain (e.g., a simulator) and must be adapted to a target domain (e.g., the real world) with only limited interactions. The authors propose DADiff, a novel framework that leverages generative models, specifically diffusion models, to capture the dynamics mismatch between domains. The core idea is to interpret the state transition as a conditional generative process and to measure the "generative trajectory deviation"—the discrepancy between the latent state trajectories of the source and target domains during the diffusion generation process. The paper provides a theoretical performance bound linking this deviation to the policy's performance gap and proposes two practical variants: DADiff-modify (which penalizes source-domain rewards based on the deviation) and DADiff-select (which filters source-domain data). The method is also extended to the Flow Matching framework. Experiments on MuJoCo environments with kinematic and morphology shifts show that DADiff outperforms several strong baselines, including PAR.

**Strengths:**

The paper offers a fresh and principled perspective on dynamics adaptation by framing it as a problem of generative trajectory mismatch. This is a significant conceptual shift from prior work that often relies on domain classifiers or single-step representation learning.

**Weaknesses:**

The primary concern is the justification for the added complexity of using a diffusion model for dynamics modeling. While the results are strong, the performance gain over the strongest baseline, PAR, is sometimes marginal (e.g., in `ant(broken hips)` or `walker(broken right foot)`). The paper acknowledges that VGDF, a model-based method, is significantly slower, but it does not provide a detailed analysis of DADiff's own computational cost (e.g., training/inference time, memory footprint) compared to PAR, which is a crucial factor for real-world applicability. The increased complexity needs a more compelling justification in terms of capability.

The experiments are conducted on standard MuJoCo locomotion tasks, which, while common, have relatively simple and deterministic dynamics. The paper’s core claim is about capturing complex dynamics mismatches via diffusion models. To truly validate the advantage of modeling the full generative trajectory, experiments on tasks with more complex, high-dimensional, or highly stochastic dynamics would be far more convincing. The current experiments, which largely follow the setup of PAR, do not fully showcase the potential benefits of the proposed method in more challenging scenarios.

**Questions:**

The paper mentions an extension to Flow Matching (Appendix C). Could the authors elaborate on the specific modifications required in the algorithm? In the diffusion framework, the deviation is calculated using the noise prediction model `ϵ_θ`. What is the direct analogue in the Flow Matching framework? Is it solely based on the vector field prediction `v_θ`, and if so, how does the continuous-time nature of the trajectory in Flow Matching affect the calculation and interpretation of the "generative trajectory deviation" compared to the discrete steps in diffusion?

---

> ### Author Response · Authors · 2025-11-22
> **Author's Response (1/2)**
>
> We are thankful for your time and help, especially related to the discussions on the justification of diffusion-based modeling complexity, extended experiments and the flow matching extension. We were glad to hear that you found our paper offers a fresh and principled perspective on dynamics adaptation by framing it as a problem of generative trajectory mismatch. To help the reviewer and AC quickly grasp our main clarifications, we summarize our responses as follows:
> - We clarify that our method maintains the same time complexity as PAR through parallel generation, with only a minor increase in memory, and we provide training time and memory footprint analyses in Figure 3 and Figure 8.
> - Our method achieves consistent improvements in 6 out of 8 tasks, and we include extended gravity/friction shift and stochastic dynamics experiments demonstrating clear robustness and improvement of our method.
> - We confirm that vector field prediction $v^\theta_\mathrm{tar}$ is directly used to compute deviation, and the process remains tractable with a limited set of predefined timesteps as detailed in Appendix C.
>
> We provide the following point-by-point responses to address the reviewer’s comments in detail.
>
>
> **[W1] Complexity concern and computational cost results**
>
> > The primary concern is the justification for the added complexity of using a diffusion model for dynamics modeling.
> >
> > The paper acknowledges that VGDF, a model-based method, is significantly slower, but it does not provide a detailed analysis of DADiff's own computational cost (e.g., training/inference time, memory footprint) compared to PAR, which is a crucial factor for real-world applicability. The increased complexity needs a more compelling justification in terms of capability.
>
> Thanks for your comments on complexity and highlighting the importance of analyzing on computational cost. We appreciate the opportunity to make a further clarification.
>
> - **Concerning complexity,** we adopt the parallel generation form when measuring the generative trajectory deviation $d(s,a,s')$, as we claim in Section 4.2. Meanwhile, we use a MLP with three linear layers to measure the deviation, which is a quite light model and is consistent with PAR. Therefore, if we ignore the complexity of the model, the time complexity of our method is $O(1)$, which remains the same as PAR. We also agree that due to the multi-step characteristic of diffusion models, the space complexity of our method is $O(K)$, where $K$ is the number of diffusion timesteps. However, compared to the iterative generation form, which would lead to an $O(K)$ time complexity, we think a slight increase in memory is much more affordable in online dynamics adaptation, which would cost much time in online interaction with environments.
> - **Concerning computational cost results, we provided a training time comparison among various methods in Figure 3 and the memory footprint comparison in Figure 8(a).** Besides, we also provided comparison results of our method across different diffusion timesteps $K$ to explore the increment of memory footprint caused by the parallel generation in Figure 8(b). Please check Figure 3 and Figure 8 for details.
>
> **[W2] Performance concerns**
>
> >While the results are strong, the performance gain over the strongest baseline, PAR, is sometimes marginal (e.g., in `ant(broken hips)` or `walker(broken right foot)`).
>
> We are glad that you found the overall results strong and that our method consistently outperforms prior approaches. Regarding the specific cases of `ant(broken hips)` and `walker(broken right foot)`, we would like to clarify that our improvements over PAR in these two settings are +637.54 (corresponding to a 19.1% improvement over PAR) and +447.1 (corresponding to a 15.2% improvement over PAR), respectively. Although the gains may appear small in Figure 2 at first glance, these improvements are meaningful. We have highlighted these values more clearly in the latest version.

---

> ### Author Response · Authors · 2025-11-22
> **Author's Response (2/2)**
>
> **[W3] Experiments on more complex tasks**
>
> >The experiments are conducted on standard MuJoCo locomotion tasks, which, while common, have relatively simple and deterministic dynamics. The paper’s core claim is about capturing complex dynamics mismatches via diffusion models. To truly validate the advantage of modeling the full generative trajectory, experiments on tasks with more complex, high-dimensional, or highly stochastic dynamics would be far more convincing. The current experiments, which largely follow the setup of PAR, do not fully showcase the potential benefits of the proposed method in more challenging scenarios.
>
> Thanks for your suggestions. To better evaluate the advantage of modeling the full generative trajectory, **we provided experiments conducted on `hopper (broken joints)` and `walker (broken right root)` with stochastic dynamics in Table 1**. We found that our method maintains robust performance even as the standard deviation $\varsigma$ increases, while PAR's performance degrades significantly. We can also compare the performance in the same task, which demonstrates that our method improves the performance up to 96.8% compared to PAR.
>
> **[Q1] Details about flow matching**
>
> > The paper mentions an extension to Flow Matching (Appendix C). Could the authors elaborate on the specific modifications required in the algorithm? In the diffusion framework, the deviation is calculated using the noise prediction model `ϵ_θ`. What is the direct analogue in the Flow Matching framework? Is it solely based on the vector field prediction `v_θ`, and if so, how does the continuous-time nature of the trajectory in Flow Matching affect the calculation and interpretation of the "generative trajectory deviation" compared to the discrete steps in diffusion?
>
> Thanks for your insightful thoughts on the extended flow matching implementation in Appendix C. We do use the vector field prediction $v^\theta_\mathrm{tar}$ directly to build the extended version. It is important that although flow matching methods contain a continuous-time nature, they still need a set of timesteps predefined by schedulers for the inference process, like `{0.0, 0.2, 0.4, 0.6, 0.8, 1.0}`, which actually has a limited amount. Therefore, we can calculate the deviation with limited and predefined timesteps. We use the average function to replace the sum function when calculating the deviation in order to avoid the influence of the total number of timesteps. We are delighted to have further discussions if needed.

---

### Meta-Review · Area_Chair_4WyF · 2025-12-31

**Summary:**

The paper makes meaningful contributions to propose a diffusion-based method to generate signals for either penalizing rewards or filtering data for policy learning under dynamics shifts. However, integrating several aspects mentioned by reviewers and my own reading, the following may help further improve the quality before it can be published: 1) limited novelty: the main idea is to generalize PAR to the stochastic settings using diffusion models' latent states; but there is a conceptual and theoretical gap between measuring (latent) trajectory distribution discrepancy and optimizing policy adaptation performance -- there is sometimes a huge variance between "-modify" and "-select" variants, which cannot be easily explained by theory presented in this paper.  2) one reviewer mentioned the "large shift" cases, including "action mask in source" type of shift and "hard" magnitudes in the benchmarks, which are extensively evaluated in previous papers -- this indicates adding more comprehensive evaluations across different types of shift is very necessary, due to the fact that dynamics shifts cover so many possible dimensions. 3) parameters for reward modification and data filtering influence the result heavily -- there is no informed way to understand the mechanism (for example, two very different parameters may achieve similar results).

**Reviewer Concerns:**

The rebuttal added many meaningful clarifications and improvements, including more variants of using the d value in the downstream policy learning, more fair comparison suggested by the reviewers, and additional shift-type and magnitude settings. However, I think there is a fundamental gap between the diffusion generative modeling and the downstream policy learning.

**Reviewer Scores:**

One reviewer indicates they will keep the score, while the other reviewers did not change the score. Given the negative reviewers are relatively confident, I predict this would be a borderline paper. So I read the paper and integrated my own opinions in the final meta-review.

---

### Decision · Program_Chairs · 2026-01-26

Reject